# Chemical Bonding of Biomolecules to the Surface of Nano-Hydroxyapatite to Enhance Its Bioactivity

Sohee Kang [1], Adnan Haider [2], Kailash Chandra Gupta [3,4], Hun Kim [5] and Innkyu Kang [4,5,*]

1 Department of Dentistry, College of Medicine, Yeungnam University, Daegu 42415, Korea; kangsh@yu.ac.kr
2 Department of Biological Sciences, National University of Medical Sciences (NUMS), Rawalpindi 46000, Pakistan; adnan_phd@outlook.com
3 Polymer Research Laboratory, Department of Chemistry, Indian Institute of Technology, Roorkee 247 667, India; kcgptfcy@iitr.ac.in
4 Department of Polymer Science and Engineering, Kyungpook National University, Daegu 41566, Korea
5 Jeil Medical Corporation, Seoul 08378, Korea; biohuny@jeilmed.co.kr
* Correspondence: ikkang@knu.ac.kr

**Abstract:** Hydroxyapatite (HA) is a significant constituent of bones or teeth and is widely used as an artificial bone graft. It is often used to replace the lost bones or in reconstructing alveolar bones before dental implantation. HA with biological functions finds its importance in orthopedic surgery and dentistry to increase the local concentration of calcium ions, which activate the growth and differentiation of mesenchymal stem cells (MSC). To make relevant use of HA in bone transplantation, the surfaces of orthopedic and dental implants are frequently coated with nanosized hydroxyapatite (nHA), but its low dispersibility and tendency to form aggregates, the purpose of the surface modification of bone implants is defeated. To overcome these drawbacks and to improve the histocompatibility of bone implants or to use nHA in therapeutic applications of implants in the treatment of bone diseases, various studies suggested the attachment of biomolecules (growth factors) or drugs through chemical bonding at the surface of nHA. The growth factors or drugs bonded physically at the surface of nHA are mostly unstable and burst released immediately. Therefore, reported studies suggested that the surface of nHA needs to be modified through the chemical bonding of biologically active molecules at the surface of bone implants such as proteins, peptides, or naturally occurring polysaccharides to prevent the aggregation of nHA and to get homogenous dispersion of nHA in solution. The role of irradiation in producing bioactive and antibacterial nHA through morphological variations in surfaces of nHA is also summarized by considering internal structures and the formation of reactive oxygen species on irradiation. This mini-review aims to highlight the importance of small molecules such as proteins, peptides, drugs, and photocatalysts in surface property modification of nHA to achieve stable, bioactive, and antibacterial nHA to act as artificial bone implants (scaffolds) in combination with biodegradable polymers.

**Keywords:** hydroxyapatite (HA); orthopedics; dentistry; chemical bonding; histocompatibility; therapeutic functions





## 1. Introduction

Hydroxyapatite (HA) has unique properties as its composition and structure are almost similar to the natural HA of human teeth and bones [1,2]. It belongs to an important class of ceramics ($Ca_{10}(PO_4)_6(OH)_2$) and is commonly used in the reconstruction of bone defects [3] resulting from trauma, congenital abnormality, and bone diseases [4,5]. It is used to repair bones due to its reasonably high bioactivity, non-immunogenic properties, biocompatibility, and bone conductivity [6]. HA is a main component of bone, and it regulates the local concentration of $Ca^{2+}$ ions that induces the proliferation of osteoblasts and promotes the growth and differentiation of mesenchymal stem cells more efficiently. Despite these advantages, the HA is also associated with some disadvantages, such as its

brittleness [7] and its nanoforms (nHA) show aggregation due to weak interface adhesion with polymers, which are significant challenges for its applications in the areas of biomedical fields, hence still it has to cover a long way to reveal its satisfying properties in bone tissue engineering [8].

HA nanoparticles of the desired size are produced by carrying out the following reaction between $Ca(NO_3)_2$ with $(NH_4)_2HPO_4$ in aqueous media within a pH range of 10–12 at a constant temperature as the size of HA particles depends on the reaction temperature.

$$10Ca(NO_3)_2 + 6(NH_4)_2HPO_4 + 8NH_4OH \rightarrow Ca_{10}(PO_4)_6(OH)_2 + 20NH_4NO_3 + 6H_2O \quad (1)$$

The main constituents of HA are Ca and P, which are in a theoretical ratio of about 1:7. To stabilize the dispersion of hydrophilic nHA with hydrophobic polymers and to increase the mechanical properties of composites, the surface of nHA is modified to control the physical or chemical interactions with polymers. The variation in the band position in typical Fourier transform infrared (FT-IR) spectra of nHA (Figure 1a) for the characteristic groups such as $PO_4^{3-}$ (1095.1, 1041.3, 962.3, 874.2, and 563.7 cm$^{-1}$) and –OH groups (3566.8 cm$^{-1}$) is frequently used to confirm the surface modification of nHA or its interactions with polymers.

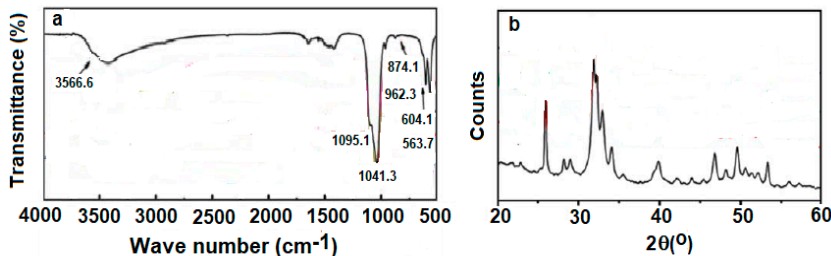

**Figure 1.** Structure characterization of nHA by FT−IR (**a**) and XRD (**b**) analyses.

Similarly, the presence of characteristic diffraction peaks in the X-ray diffraction (XRD) spectrum (Figure 1b) of nHA at 25.87° (002), 31.81° (211), 32.17° (112), 32.86° (300), 34.04° (202), 39.85° (310), 46.74° (222), 49.50° (213) and 53.24° (004) are found useful to confirm the crystalline structures of nHA. The appearance of broad peaks in XRD spectra is due to nanosize structures of nHA. The presence of these peaks in the XRD spectrum suggests hexagonal crystalline lattice in nHA particles (JCPDS 09-0432). The nHA crystals are composed of six $PO_4^{3-}$ groups and ten calcium atoms, in which oxygen and calcium atoms have three and two unique environments, respectively [9]. The nHA with 1:7 stoichiometry has a hexagonal lattice system with two major planes, i.e., plane a and plane c. Plane c is rich in phosphate ions ($PO_4^{3-}$) and hydroxyl ($^-OH$) ions, hence considered a negatively charged plane, whereas plane a is rich with calcium ($Ca^{2+}$) ions, hence considered a positively charged plane. This anisotropic nature of nHA is considered responsible for the anisotropic adsorption of biomolecules [10]. The area of the a- and c-planes are different; hence, they have different capacities to adsorb biomolecules besides showing the variation within the pH and ion composition of the media. HA accept varieties of ions along the c-plane and show variation in band gap depending on the size of the ions and particularly on interactions of biomolecules in a biological system. As HA is osteoconductive (bioactive), hence it induces growth when placed in the vicinity of viable bone or differentiated bone-forming cells. The surface properties of HA and its biocompatibility are modified with ion substitutions. For example, strontium substitution in HA produces a positive effect on alkaline phosphatase activity of seeded osteoblasts. The modification of HA surfaces with organic molecules produces morphological changes and also provides opportunities for chemical bonding with biomolecules.

In orthopedic surgery, autografts are considered the gold standard, but they are associated with various drawbacks, such as donor site morbidity due to infection and hematoma formation. To overcome these drawbacks of autografts and to reduce the

potential risk of infections, efforts have been made to develop mimetic scaffolds to act as an extracellular matrix (ECM) to repair bone defects in combination with biodegradable synthetic polymers such as poly (lactic acid) (PLA) [11], poly (glycolic acid) (PGA), poly (lactic-co-glycolic acid) (PLGA) [12–14] and natural polysaccharides such as chitosan (CS) for bone regeneration. The combination of HA with biodegradable polymers not only increases the chemical properties of resultant composite scaffolds but also increases their affinity to protein adsorption. In only polymer-based scaffolds, the osteoblasts are attached to the outer surfaces of polymer scaffolds, but in polymer composite scaffolds having nHA, the osteoblasts are able to attach more efficiently and show enhanced alkaline phosphatase activity (ALP), i.e., the expression of bone-specific markers (coding bone sialoprotein and osteocalcin). The ALP enzyme acts as a marker to ensure the osteogenic differentiation of bone-forming cells; hence the increase in ALP activity of cells is indicative of bone formation. The ALP activity is measured by recording the optical density ($\lambda = 405$ nm) of p-nitrophosphate (pNP) produced by the hydrolysis of p-nitrophenylphosphate ester (pNPP) by the ALP enzyme in alkaline buffer solution and using the following relationship:

$$\text{ALP activity (mU / mL)} = A / (V.T) \qquad (2)$$

where A is the optical density of pNP (nmol), V is the volume of assay well (mL), and T is time (minutes). Thus, an increase in ALP activity on seeding cells on the polymer–HA composite scaffolds is a direct measure of osteogenic properties of scaffolds, which is usually found to be high on adding HA in polymer matrices.

It is believed that the composite scaffolds of polymers and nHA help in the promotion of cell adhesion and growth of osteoblasts and are able to stimulate the functional activities of bone cells. To further improve the activities of polymer–nHA composite scaffolds, the blending of biological proteins (such as collagen) or growth factors (such as bone morphogenetic protein-2 (BMP-2)) was found to be useful. Though HA is used widely as an osteoconductive substitute for bone or coating materials to induce bioactivity in orthopedic and dental implants but without having cell-adhesive proteins at its surface, significantly high osteoconductivity of HA is not achieved at the site of its application.

The surface coating of nHA on implants modified with biological substances such as ECMs or arginine–glycine–asparagine (RGD) peptide improves cell adhesion, proliferation, and differentiation, which are essential steps in bone formation. In vitro studies of the surface modification of HA with RGD showed a significant improvement in the adhesion of osteoblasts [15]. In the applications of nHA in bioactive scaffolds, one of the following three strategies is used to modify the surface of nHA by varying the relative ratio of the nHA and the employed protein- or peptide-coating materials:

(a) The coating may produce morphological changes at the surfaces nHA (Figure 2a) that will influence surface roughness, which improves the mechanical properties of bone and help in stimulating the activity of cells;

(b) The coating may produce a compositional variation in surfaces of nHA (Figure 2b) by creating functional groups chemically [16] or through irradiation, which helps in promoting bone formation at the surface of implants (scaffolds). This strategy may be called bioactive coatings;

(c) The coating may produce biological functionalities by adding biomolecules (Figure 2c) to stimulate cell adhesion and proliferation for bone tissue formation. For example, the activity of osteoblasts was high in scaffolds having RGD peptide-coated nHA [15].

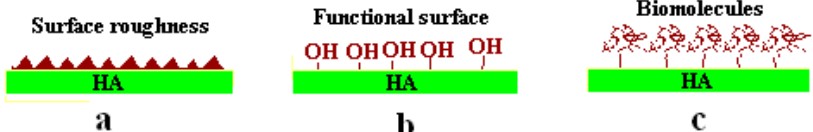

**Figure 2.** Strategies for surface modifications of nHA by (**a**) roughness; (**b**) functionality; (**c**) deposition of biomolecules.

The polymer–nHA composite scaffolds have shown high activity; hence, they act as potential bone graft substitutes in reconstructive surgery [17]. As the compressive strength of pure PLGA is not sufficient, hence reinforcement of nHA in PLGA has shown a significant improvement in the mechanical properties of PLGA-based scaffolds. The nHA/PLGA composite scaffolds show increased cell proliferation, ALP activity, and osteogenic properties in comparison to scaffolds fabricated with pure PLGA. In nHA/PLGA scaffolds, the PLGA also shows a reduced rate of its biodegradation. The presence of nHA in composite scaffolds also controls the medium pH that creates a suitable microenvironment for the attachment and growth of bone cells.

The hydrophobic synthetic polymers, such as PLGAs, find it difficult to interface easily with hydrophilic nHA; hence, to improve the compatibility of nHA with hydrophobic polymers, the surface of nHA is modified chemically using small molecules or hydrophilic polymers. In this direction, the properties of a coupling agent, such as hexamethylene diisocyanate, are utilized for the chemical binding of commonly used polymers such as polyethylene glycol (PEG) [18] or polycaprolactone (PCL) [19] at the surfaces of nHA. However, the application of isocyanates as a spacer between PEG and nHA is found to be less useful due to its poor biocompatibility. From a chemical point of view, the interfacial interactions between the polymer and inorganic framework may be strong (covalent, coordination, ionic), weak (van der Waals, hydrogen bonding, hydrophilic–hydrophobic balance), or no interactions. Since the interfacial properties between polymer and nHA greatly depend on the chemical modification of nHA, Li et al. [20] attached stearic acid (Sa) to the surface of nHA to improve the interfacial interaction between nHA and polymer. The chemically modified nHA with bioactive molecules is dispersed homogeneously in solution, and resultant polymers composite scaffolds show enhanced bioactivity [21]. In order to improve the interaction and compatibility of nHA with hydrophobic polymers, another strategy has been recently used to commercialize HA by dispersing it in biodegradable poly($\varepsilon$-caprolactone) using a copolymer with small phosphate units [22].

The surface modification of nHA basically influences the work of adhesion between the polymer and nHA, which has been found to vary between 48 $Jm^{-2}$ for poly(caprolactone) and 63 $Jm^{-2}$ for polylactide with nHA and with a contact angle $< 60°$. The nHA-polymer composite scaffolds with low contact angles have wettable surfaces, which is suitable for the ingrowth of bone cells (osteoblasts). The properties of polymer–nHA composites are influenced by the shape, size, and size distribution of nHA as well as with the properties of the polymer, such as the molecular weight of polymers (Mc) and their interface with added nHA. The physical properties of nHA, such as size and size distribution, play significant roles in controlling the mechanical properties of the composite scaffolds, such as the modulus of elasticity ($E_\infty$), as observed in the case of interpenetration poly (titanium oxide) and poly (hydroxyl ethylmethacrylate) composites [23].

$$M_c = \frac{3\rho RT}{E_\infty} \qquad (3)$$

where $\rho$ is polymer density; R and T are gas constant and absolute temperature. $E_\infty$ is the equilibrium elastic modulus of polymerHA composite. The natural bones have nanosized blade crystals from the HA in close contact with collagen fibers.

Currently, attention is also paid to ascertain if the irradiation of HA can cause variation in its bioactivity and its other properties by using high-energy radiation such as UV, X-rays, or electron beam irradiation. On this concept, Baltacis et al. [24] studied the effect of structural point defects on the surface properties of HA. They used threshold photoelectron emission spectroscopy and measured the work function of HA as well as of its biological samples. To predict the effect of radiation on the surface properties of HA, density functional theory (DFT) and other methods of approximations were used to calculate the effect of structural defects that might be produced in HA lattice on irradiation. The calculations have shown that the generation of point defects and deposition of charge at the surface of HA were due to the variation in the spatial arrangement of atoms in the

lattice of HA and the creation of vacancies, namely $PO_4{}^{3-}$, OH, and oxygen in HA. These variations in surface properties of HA influence the interactions of HA with osteoblasts leading to show enhanced bone tissue formation [25,26], and these interactions are based on theoretical calculations. However, information on the direct effect of in vivo radiation effect on interactions between enamel and organic matter is not explicitly available. Madrid et al. [27] evaluated the macroscopic, microscopic, microstructural, and morphological aspects of human teeth enamel extracted from head and neck radiotherapy of teeth and compared with enamel collected from non-irradiated teeth of the patients. Null hypothesis testing has been suggested in the absence of any micro-morphological difference between in vivo irradiated and non-irradiated tooth enamel. Most of the reported studies indicated that the function of peptides on the surface of HA depends on their folded structures and conformations, which are stabilized by a specific microenvironment present at the surface of HA due to the presence of various types of ions. The biomimetic activity of peptides on the re-mineralization of human dentine has shown significant variation in the folded structures of peptides [27].

Thus, in this review, we attempted to introduce the role of polymer-based implants (scaffolds) with improved bioactivity. We also highlight the contributions of physically or chemically bonded biomolecules, such as proteins or drugs at the surface of nHA, to produce bioactive composite scaffolds by using biodegradable polymers for bone tissue engineering. Efforts are also made to discuss the improved cell compatibility and properties of drug-loaded therapeutic scaffolds [28] by using chemically modified nHA and biodegradable polymers such as PLGA.

## 2. Bioactivity of Scaffolds Having Nanohydroxyapatite Particles Modified with Natural Molecules or Drugs

The dispersion, stability, and bioactivity of hydroxyapatite nanoparticles in polymer-based scaffolds have shown significant variations in using nHA particles attached physically or chemically with biomolecules or drugs at their surfaces. As nHA particles having physically adsorbed molecules at their surfaces have shown poor stabilities of these species at their surfaces; hence the effect of the nHA particles modified chemically by biomolecules or drugs at their surface is discussed in detail in terms of controlling the bioactivity of polymer scaffolds, which are used in bone tissue formation.

### 2.1. Bioactivity of nHA Particles Modified Chemically with Natural Molecules

HA is a crystalline inorganic material with excellent biocompatibility and is commonly used in the fabrication of orthopedic and dental implants (scaffolds) in combination with various biodegradable polymers. The blending of the nHA particles in the polymer scaffolds increased the effect on the mechanical and biological activity of scaffolds used in bone formation [29]. However, it is commonly observed that nHA particles become agglomerated in solutions of various polymers such as poly (lactic acid), which is used in the fabrication of bone implants. To prevent the agglomeration and precipitation of nHA, the nHA was stabilized in an aqueous spinning solution containing sodium alginate. Then, the composite bioactive fiber membranes were electrospun, which were homogenously distributed in nHA particles in poly (vinyl alcohol) composite fiber membranes [30].

These studies indicated that the addition of the nHA particles in such polymers remained ineffective in increasing the mechanical strength of the implants unless there were strong interactions between the nHA particles and polymer matrices. The poly (lactic acid) is a representative biodegradable polyester, but the dispersion of the nHA particles in PLA is found to be difficult. To overcome the difficulties of the nHA particles, Zhao et al. [31] modified the surfaces of the nHA particles through the chemical bonding of naturally occurring epoxidized soybean oil by a melt-blending process, then 20 wt% of modified nHA particles were mixed in PLA. The resultant PLA composites containing surface-modified nHA particles with 20 wt% of epoxidized soybean oil were able to maintain a flexural strength of 71.6 MPa. The number of cells cultured at the surface of these composite

scaffolds (nHA/PLA) was found to be 10% higher than what was observed on pure PLA scaffolds using osteoblasts in the same experimental conditions.

Similarly, Jiang et al. [32] reported in their studies that chemically modified nHA particles with citric acid were able to disperse efficiently in PLGA solution and produced scaffolds with improved mechanical strength, which is required in bone implant applications. The homogenous dispersion of chemically modified nHA particles was found to be possibly useful in the fixation of intra-fracture bone defects more efficiently. These investigations clearly revealed the importance of naturally occurring small molecules, such as soyabean oil or citric acid, in the surface modification of nHA and in developing bioactive scaffolds for bone implant applications.

### 2.2. Drug-Releasing Orthopedic Implants Having Chemically Bonded Drugs at the Surface of nHA

Orthopedic implants are also used as drug-releasing devices mainly to target bone infections (osteomyelitis). These implants are basically a combination of ceramics and antibiotics. Due to the poor accessibility of sites of infection to systematically administered antibiotics, the application of drug-releasing implants of bioresorbable ceramics loaded with antibiotics is considered the best substitute. However, the efficacy of such drug-releasing implants depends on the properties and structures of bioresorbable calcium phosphate used in implant formulations. The chemically bound drug from these implants is released at the site of infection for several months reasonably at a slow rate. This approach to implant fabrication is also found useful for associating various growth factors such as transformation growth factors, insulin-like growth factors, and bone morphogenetic proteins as well.

To utilize the advantages of pores and surface area of nHA in the fabrication of implants with increased capacity for drug-loading, Yu et al. [33] fabricated highly porous nHA microspheres by using a novel ice-templated spray drying (ITSD) technique and by using different amounts of polyvinylalcohol (PVA, Figure 3). The loading of gentamicin sulfate in these porous nHA microspheres was carried out by dispersing porous nHA microspheres in the solution of gentamicin sulfate. In vitro drug-release studies indicated that loaded drug was released within an initial period of 18 h due to the presence of large-sized pores (~2.5 μm) in nHA microspheres and due to physically loading of drug in porous nHA microspheres rather than chemical bonding. These studies identified the gap and suggested clearly to prolong the slow release of the drug and to improve the efficacy of the drug, the drug need to be loaded through a chemical bonding even if the pore size is small.

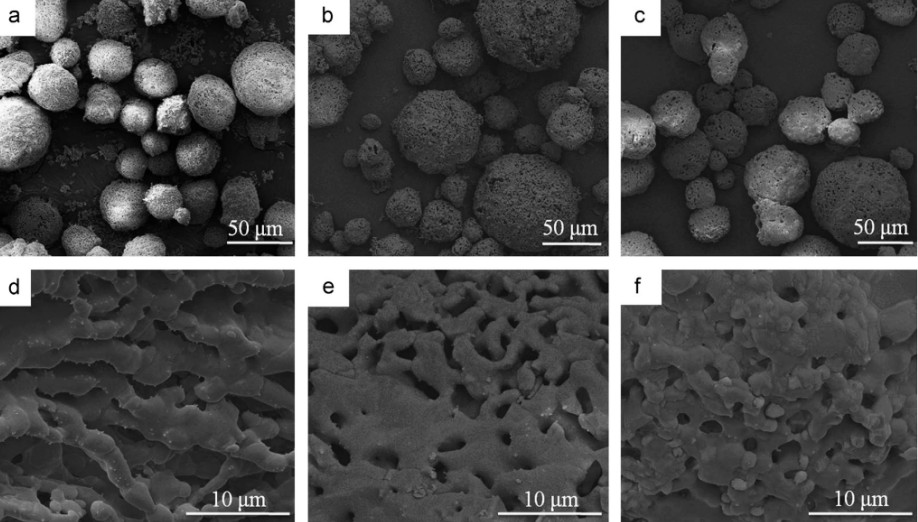

**Figure 3.** SEM micrographs of porous HA microspheres fabricated from suspensions with different PVA contents, (**a**,**d**) 0.5 wt%, (**b**,**e**) 1.0 wt%, and (**c**,**f**) 1.5 wt%. (**d–f**) at low and high magnification. Reprinted with permission from Ref. [33]. Copyright (2022) MDPI AG.

In some studies, the loading of drugs on nHA particles was carried out differently by simultaneous co-precipitation of the nHA particles and loading of the drug during the formation of the nHA particles [34,35]. For example, doxorubicin (DOX) was mixed with an aqueous solution of $Ca(NO_3)_2$, and then it was added slowly to an aqueous solution of $(NH_4)_2HPO_4$ under vigorous stirring to form drug-loaded nHA particles. Considering the drawbacks of the fast release of the loaded drug, the surface of the nHA particles was functionalized with a silane coupling agent and folic acid (FA) to bind the loaded drug chemically with the nHA particles. The loading of folic acid on nHA particles was specifically carried out to add cancer cell targeting properties in nHA particles (Figure 4). The in vitro release of loaded drugs from nHA particles was studied by dispersing drug-loaded nHA particles in a weakly acidic buffer solution, which has shown a controlled release of DOX from nHA particles (DOX-nHA-FA) within a period of 24 h. In addition to this, the prepared nHA particles also showed enhanced cellular uptake by cancer cells through folate receptor-mediated endocytosis that ultimately inhibited the proliferation of target cancer cells more efficiently.

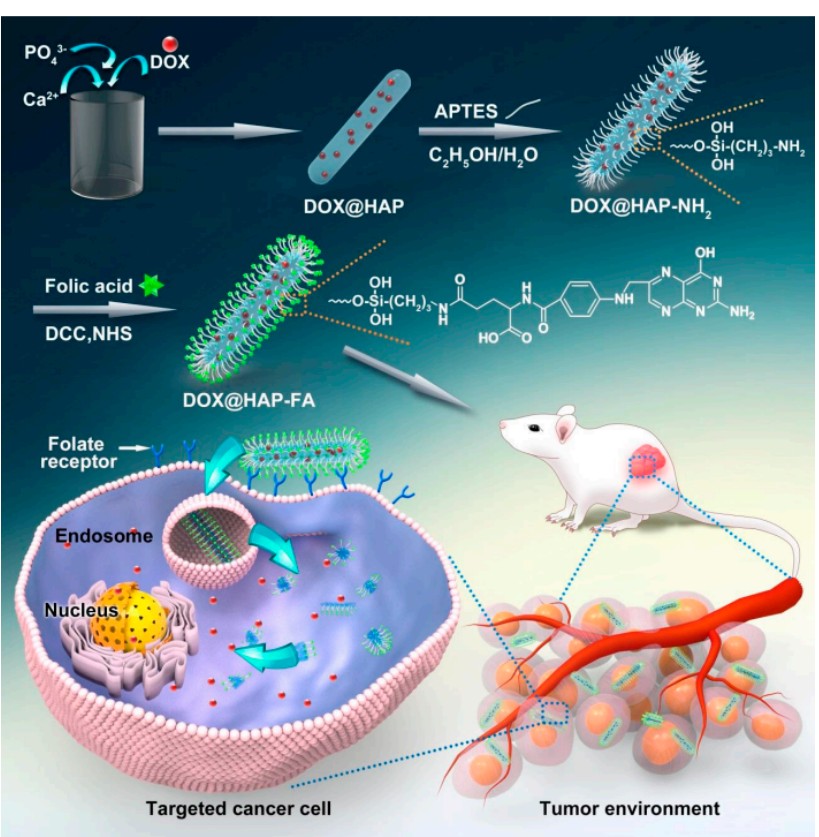

**Figure 4.** Synthesis route of DOX@HAP-FA nanocarrier and the antitumor mechanism of DOX@HAP-FA in cancer cells. Reprinted with permission from Ref. [34]. Copyright (2018) American Chemical Society.

To improve the efficacy of the drug used commonly in the treatment of osteoporosis, the PLGA scaffolds were prepared [28] by dispersing nHA particles having chemically bound pamidronic acid (Pa). These PLGA scaffolds were able to reduce the hyperactivity of the osteoclast and maintained an optimum balance between the formation and resorption of bones during the process of bone tissue formation. In these investigations, L-glutamic acid was first bonded chemically at the surface of the nHA particles to create carboxyl groups at the surface of the nHA particles, then allowed to react with the pamidronic acid (Figure 5).

**Figure 5.** Schematic diagram depicting the grafting of pamidronic acid at the surface of the nHA particles. Reprinted from Ref. [28] with copyright (2012) permission from the Royal Society of Chemistry.

The pamidronic acid-modified nHA particles (Pa-g-nHA) were electrospun as nanofiber scaffolds (Pa-g-nHA/PLGA) by using Pa-g-nHA dispersion in a solution of PLGA in chloroform. The FT-IR analysis of nHA, Pa-g-nHA, PLGA, and Pa-g-nHA/PLGA nanofiber scaffolds confirmed the successful loading of pamidronic acid in nanofiber scaffolds (Pa-g-nHA/PLGA) and also confirmed the chemical bonding of pamidronic acid with nHA particles through L-glutamic acid as a spacer (Figure 6). FT-IR spectrum of the nHA particles has shown two characteristic sharp bands corresponding to tetrahedral phosphate groups ($PO_4$) in the frequency region of 1000–1100 $cm^{-1}$ and a band at 3580 $cm^{-1}$ corresponding to free hydroxyl groups Figure 6a The spectrum of PLGA nanofibers has shown a sharp band for the carbonyl group (>C=O) at 1720 $cm^{-1}$ and two bands for hydrocarbon groups (>CH–, >CH$_2$) between 2950–3000 $cm^{-1}$ Figure 6c. The appearance of absorption bands corresponding to amide I and II bonds at 1648 and 1540 $cm^{-1}$ confirmed the chemical grafting of pamidronic acid at the surface of the nHA particles through L-glutamic acid used as a spacer. The spectrum of Pa-g-nHA/PLGA (Figure 6d) has shown all characteristic bands of nHA and PLGA. The weak bands at 1648 $cm^{-1}$ and 1540 $cm^{-1}$ were due to stretching vibrations of amide I (–CONH–) and bending vibrations of amide II (–CONH–), respectively. These results confirmed that the pamidronic acid-modified nHA particles (Pa-nHA) were well dispersed in PLGA nanofiber scaffolds. The potential of Pa-g-nHA/PLGA scaffolds as implants for treatment of osteoporosis was evaluated by conducting in vitro experiment using osteoclasts and osteoblasts. The viability of osteoclasts responsible for bone resorption was found to be reduced significantly on using Pa-g-nHA/PLGA scaffolds (Figure 7b), but osteoblasts have shown increased proliferation with Pa-g-nHA/PLGA scaffolds as compared to nHA/PLGA and pure PLGA nanofiber scaffolds, which was confirmed on comparing the fluorescence images of these scaffolds (Figure 7). These studies also suggested that chemically attached pamidronic acid in the nanofiber scaffold, Pa-g-nHA/PLGA, was nontoxic, and the scaffolds were able to act as controlled drug-releasing implants for the treatment of osteoporosis as well as for other bone diseases.

Currently, several studies are conducted to control the drug release and toxicity of drugs using nHA particles as drug-release carriers. However, the loading of drugs on nHA particles is found to be difficult due to the incompatibility of nHA with drugs, which are mostly organic in nature. Therefore, to make nHA particles compatible with drugs, two methods are commonly used to increase the loading efficiency of drugs on nHA. In the first method, the loading of the drug on nHA particles is increased by increasing the extent of physical adsorption of drugs at the surface of nHA by using porous nHA particles with a high surface area [33,34]. In the second method, the reactivity of nHA with a drug is

increased by surface modification of the nHA particle with certain biomolecules or organic molecules. The second method was found to be efficient in producing drug-releasing implants due to the enhanced affinity of the nHA particles with organic drugs as well as in increasing the dispersion of the nHA particles in hydrophobic polymers such as PLGA used in the fabrication of scaffolds.

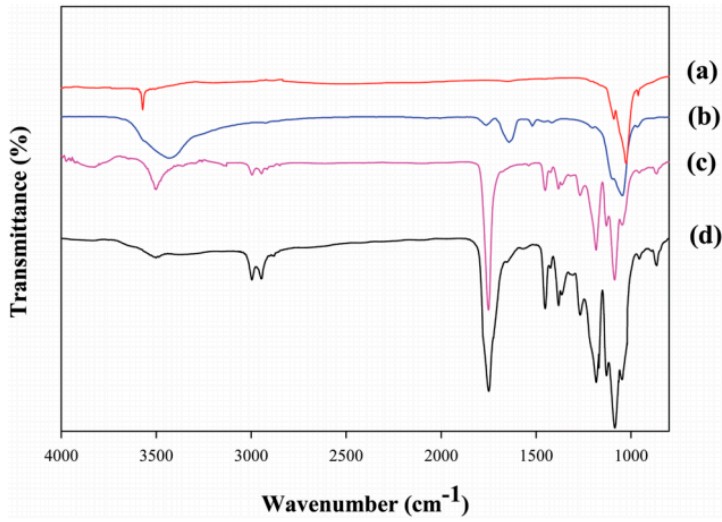

**Figure 6.** FT−IR spectra of (**a**) nHA, (**b**) Pa−g−nHA, (**c**) pristine PLGA, and (**d**) Pa−g−nHA/PLGA. Reprinted from Ref. [28] with copyright (2012) permission from the Royal Society of Chemistry.

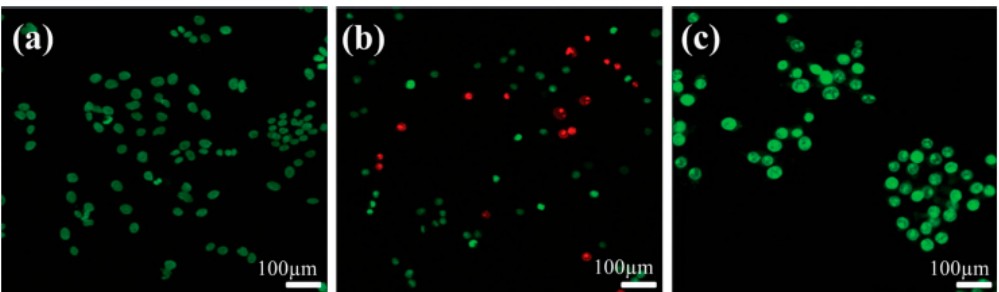

**Figure 7.** Fluorescence microscopic images of (**a**) osteoblast, (**b**) osteoclast, and (**c**) macrophage cells cultured on Pa-g-nHA/PLGA nanofiber scaffolds. Cells were stained with calcein-AM (green) and propidium iodide (red). Reprinted from Ref. [28] with copyright (2012) permission from the Royal Society of Chemistry.

To improve the dispersion of the nHA particles, Zhang et al. [35] prepared drug-releasing nHA-pamidronate particles using polyoxyethylene stearyl ether nonionic surfactant (Pa-Brij-78) in the presence of the nHA particles having chemically bonded pamidronic acid. In these studies, three types of nHA particles functionalized with Pa were prepared viz: (i) solid lipid (Pa-SNP), (ii) nanoemulsion (Pa-NEM), and (iii) PLGA (Pa-PNP) nanoparticles. The Pa-modified nHA particles were loaded with curcumin as a model drug. However, the preparation of Pa-modified drug-releasing nHA particles, including the synthesis of Pa-Brij-78, was found to be a complex process and required significant modifications in surface properties scaffolds to develop efficient drug-releasing implants for the treatment of bone diseases.

In another study, Shin et al. [36] prepared nanofiber scaffolds with an osteoporotic drug (pamidronic acid) attached with titania ($TiO_2$) and nHA particles through APTES and PEG as coupling agents. For this purpose, the nanofiber scaffolds were electrospun using a solution of $TiO_2$ and nHA particles. After that, the silane coupling agent, APTES, and PEG were attached sequentially at the surface of scaffolds for chemical bonding of

pamidronic acid (Figure 8). The bioactivity of $TiO_2/nHA$ composite nanofiber scaffolds was evaluated by culturing the differentiated macrophages (osteoclast) for 3 days. After that, the cells were stained with calcein-AM and propidium iodide to analyze the live and dead cells from green and red fluorescence images (Figure 9a). As differentiated macrophages have shown only green fluorescence, hence it indicated that $TiO_2/nHA$ composite nanofiber scaffolds were able to protect macrophases (osteoclast) as live, but cells seeded on $TiO_2/nHA$-Pa composite nanofiber scaffolds have shown red fluorescence. This has suggested that osteoclasts were killed by scaffolds loaded with pamidronic acid (Figure 9a). These studies also suggested that chemically bound Pa with nHA was more effective in the treatment of osteoporosis due to the slow and controlled release of loaded Pa from composite nanofiber scaffolds. Thus, it is also presumed that these formulations may be used further to develop oral delivery of Pa without any side effects. The chemically attached Pa on nHA has helped in increasing the efficacy of the drug due to the slow and controlled hydrolysis of nHA attached drug in comparison to unbound or physically bound Pa with nHA.

**Figure 8.** Schematic diagram showing the immobilization of pamidronic acid using a silane coupling agent and PEG on a nanofiber mat composed of Titania and HA ref. [36].

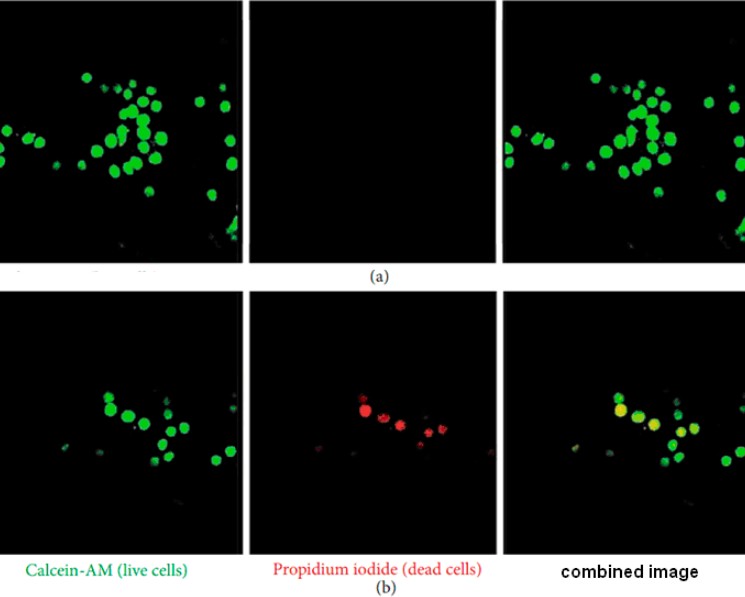

Calcein-AM (live cells)  Propidium iodide (dead cells)  combined image
(b)

**Figure 9.** Fluorescence microscopic images of calcein-AM (green) and propidium iodide (red) staining of differentiated macrophages on (**a**) $TiO_2/HA$, and (**b**) $TiO_2/HA$-P scaffolds after seeding for three days (magnification: ×200) Ref. [36].

Considering the advantages of the chemical bonding of drugs with nHA, some studies were carried out by using bisphosphonate as a potential drug for osteoporosis. Studies indicated that the loading of bisphosphonate in calcium orthophosphate (CaP) is able to show an improved release of bisphosphonate without any secondary side effects, such as nephrotoxicity as observed commonly in systematic treatments of bone diseases like osteoporosis. The osteoarticular disorders associated with increased osteoclast activity in bone resorption are treated by using bisphosphonates attached chemically with hydroxyapatite as a potent inhibitor to osteoclasts. The bisphosphonates bonded chemically with nHA are found to be more efficient and show reduced nephrotoxicity. The bisphosphonates [37,38], including ibandronate [39], pamidronate [28,36], and zoledronate [40–42], are commonly used as useful potential drugs to prevent the loss of bone density in the treatment of osteoporosis and other similar bone diseases. The orthopedic drug delivery implants with HA and zoledronate are the latest generation of bisphosphonate, which are more effective in orthopedic applications. Various types of bisphosphonates used with HA in the fabrication of orthopedic drug-release implants for various applications are listed in Table 1.

**Table 1.** Bisphosphonates in implant applications.

| Bisphosphonates | Carrier | Applications | References |
|---|---|---|---|
| Zoledronate | HA | Osteoporotic bone around implants | Peter et al. [40] |
| -do- | -do- | Bone resorption around osteoporotic implants | Tanzer et al. [41] |
| -do- | -do- | -do- | Peter et al. [41] |
| -do- | -do- | -do- | Roussie et al. [38] |
| Ibandronate | -do- | -do- | Kurth et al. [39] |
| (3-Dimethylamino-1-hydroxypropylidene)-1,1-P-C-P | -do- | Alveolar bone destruction in periodontal diseases | Denisen et al. [37] |
| Pamidronate | -do- | Osteoporosis | Haider et al. [28], Shin et al. [26] |

These studies indicated that orthopedic drug-releasing implants are able to induce faster bone healing, as well as provide a more mechanically stable situation for fixation of implants. As gelatin contributes significantly to bone regeneration [43] due to its close compatibility with the physiological composition of bones, gelatin-based scaffolds using nHA were also prepared for bone regeneration and to repair osteochondral defects [44].

## 3. Bioactivity of Scaffolds Having Small Molecules/Proteins-Modified nHA

The growth of hydroxyapatite during bone formation is closely associated with the extracellular matrix environment and possibilities of mineralization—it occurs at specific sites of HA particles under physiological conditions. In bones and dentins, the nucleation of calcium phosphate occurs within a matrix of self-assembled collagen molecules in the form of a pseudohexagonal array. The growth of HA at its specific place and in specific shapes is controlled by bonding with growth-modulating small molecules on specific faces of HA crystals [45], which limit the growth rates in specific directions to produce plate-like HA crystals (Figure 10). The resultant changes observed in surface morphology of HA were induced by stronger bonding of charged small molecules or peptides at (100) than (001) faces of HA comparable to chemical bonding. The bonding effect of small molecules in regulating the growth of HA crystals [46] also increased with the increase in the magnitude of the charge on small molecules, such as observed in the case of bone sialoprotein (BSP), osteopontin (OPN), dentin sialophosphoprotein (DSPP), dentin phosphoprotein (DPP), and amino acids [45,47]. The charge-dependent growth of HA also showed variation with the concentration of small molecules (50–80 mM). However, peptides can slightly influence the growth of HA at high concentration (>80 mM) than small molecules. The

morphological changes observed in HA in the presence of peptides were almost similar to those as observed in dentin in the presence of motif-programmed artificial peptides [48].

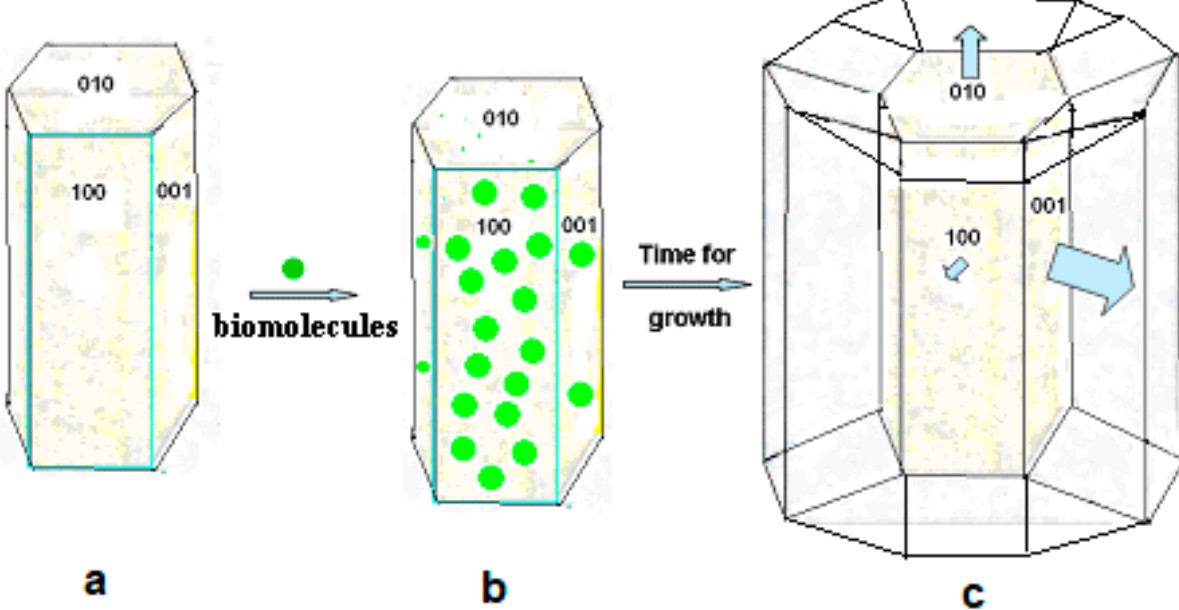

**Figure 10.** Schematic representation of HA growth on the deposition of small molecules at (100); (010), and (001) planes of HA. (**a**) Neat HA, (**b**) HA with deposited small molecules at different surfaces and (**c**) showing the formation of plate-like HA crystal. Reprinted with permission from Ref [45]. Copyright (2015) Royal Society of Chemistry.

The inorganic phosphate ($PO_4^{3-}$) concentration at 100 and 001 surface of HA remains independent of the solution's pH, such as 1.44 anions $nm^{-2}$ and 3.73 anions $nm^{-2}$, respectively, but it is $Ca^{2+}$ ions (Table 2) which showed variation with the solution's pH and controlled the growth of HA during bone formation [49].

**Table 2.** pH-dependent amount of $Ca^{2+}$ and $PO_4^{3-}$ at various planes of HA [49].

| [Ions] at Plane | Solution pH | | | | |
|---|---|---|---|---|---|
| | **4.5** | **7.4** | **9.0** | **12.3** | **14.0** |
| $Ca^{2+}$ $nm^{-2}$ (100) | 2.16 | 2.58 | 2.88 | 3.25 | 3.61 |
| $Ca^{2+}$ $nm^{-2}$ (001) | 0.37 | 1.79 | 2.48 | 3.41 | 4.35 |
| $PO_4^{3-}$ $nm^{-2}$ (100) | 1.44 | 1.44 | 1.44 | 1.44 | 1.44 |
| $PO_4^{3-}$ $nm^{-2}$ (001) | 3.45 | 3.73 | 3.73 | 3.73 | 3.73 |

The predominant crystal face of HA is (100), which gets oriented along the length of collagen fibril due to its high surface area in comparison to (001) and (010) faces of HA. The attachment of biomolecules at HA interface is of vital importance to understanding the interactions between HA and organics responsible for the biologically directed formation of bones. It is also well reported that HA surface geometry is significantly influenced by the interactions of living cells [46,50] and the pH variation in the presence of citrate ions [51,52]. For example, in vitro sialo bone protein influences the formation and growth of HA in a particular direction. In view of these considerations, the bonding of small biological molecules at the surface of HA is relevant to controlling the growth of HA during bone formation. The chemical bonding of biomolecules at the surface of HA, such as BMP-2, peptides, and vascular endothelial growth factor [45], influences the regeneration of bones more effectively. They can change the morphology of macrophages by providing different immune environments and inducing the recruitment and differentiation of osteoblasts.

The bonding of proteins such as fibronectin (Fn) and laminin (Ln) at the surface of HA enhances the adhesion and spreading of preosteoblasts, which contribute to the osteoconductivity of HA. The application of the α-chain of laminin promotes the adhesion of odontoblasts and differentiation. The laminin with a special sequence induces faster osseointegration in vivo [53]. However, there are limited studies on the bonding of Ln at the surface of bioactive HA and its cell-adhesion properties [54]. The strong interactions between HA and laminin are comparable to chemical bonding, as explained by considering the value of the Hill constant (>1). One of the positively charged faces of HA, i.e., the $Ca^{2+}$ ions-containing plane, has strong interactions with negatively charged laminin to effect the strong bonding between HA and laminin [55].

The non-collagenous family of small proteins (NCP) consists of four members, OPN, BSP, dentin matrix protein 1 (DMP1), and dentin sialophosphoprotein. These proteins have some common characteristics, including a collagen-binding domain, the HA-binding domain, and a cell-binding arginine–glycine–aspartic acid sequence. All these proteins are acidic and contain a high degree of random coil structures. The bonding of these proteins with HA plays an important role in bone formation by enhanced cell adhesion, nucleation of minerals and their maturation. The DPP and DSP are the cleaved fragments of dentin sialophosphoprotein that contain a unique sequence of aspartic acid–serine–serine (DSS) amino acids, which are responsible for the binding of calcium and subsequent mineralization and growth of HA. This indicates the potential role of these proteins in biomineralization of collagen during bone formation. Additionally, it has been observed that collagen–BSP interactions also promote the formation of HA as reported in several in vitro studies [56,57].

To enhance the amount of small bioactive molecules at the surface of HA, efforts were also made to utilize the properties of nanoporous HA. The macro- and microscale pores in HA enhance cell adhesion and biomineralization. However, the biological effects of nanopores in HA with sizes in the tens of nanometers range, similar to those in natural bones, are rarely studied. The physical adsorption of small biomolecules such as amino acids [48,58,59], citrate [52,60], and numerous other molecules regulates nucleation/growth and modulates the size and morphology of HA crystallites, which is well established from X-ray diffraction data and electron microscopy of HA bonded with amino acids [61,62]. However, very few experimental techniques can probe the chemical binding of small molecules at the surface of HA, and most of the current insights available for protein/peptide, amino acids, and citrate binding at the surface of HA and bone minerals stem from solid-state nuclear magnetic resonance (NMR) analysis [45,63,64]. The $^{13}C\{^{1}H\}$ heteronuclear correlation (HETCOR) spectra of L-glutamic acid (Glu) and O-phospho-L-Serine (Ser-$OPO_3$) molecules bonded at the surface of HA (Figure 11a,b) provide information about the interactions between the C atoms of the bonded amino acid with H atoms on HA or with H atoms of amino acid [45].

For the Glu-modified HA (Figure 11a), the major correlation of $^{1}H$ peaks appeared at 0 ppm, which corresponds to the hydroxyl group in HA, suggesting interactions between Glu and HA. The $^{13}C$ correlation peak that appeared at ~180 ppm has indicated that carboxyl groups in Glu have interactions with HA (Figure 11a). For Ser-$OPO_3$ (Figure 11b), the broad correlation peaks observed of $^{1}H$ at 0 ppm were due to the contribution of protons both in amino acids and HA.

The appearance of $^{1}H$ peaks near 0 ppm and $^{13}C$ at 170 ppm indicated that for the possible interactions between the carbon atoms in Ser-$OPO_3$ and protons present at the surfaces of HA (Figure 11b). These results suggested that amino acids interacted with surfaces of HA surfaces and were interacting with most of the protons present at the outermost surface of HA. Thus, these NMR data were able to confirm the interactions between HA and amino acids used for the modification of surface properties of HA, as other techniques are not able to provide information on these interactions.

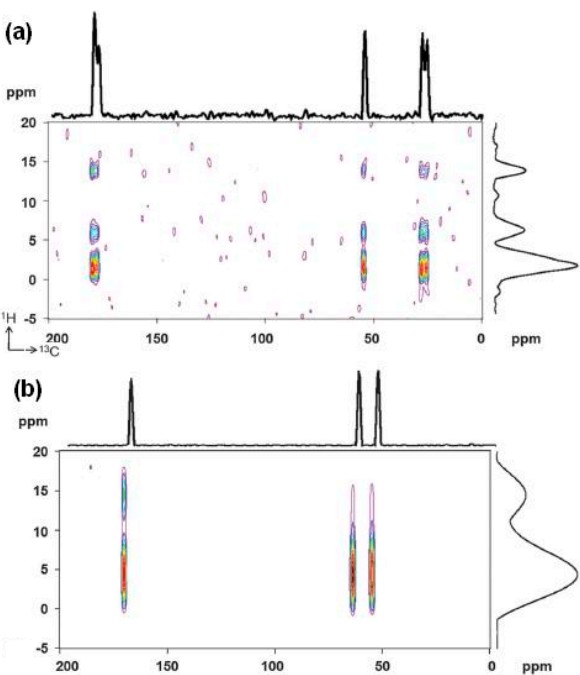

**Figure 11.** $^{13}$C{$^1$H} HETCOR 2D spectrum of (**a**) 200 mM Glu and (**b**) 200 mM Ser-OPO$_3$. From Ref. [45] with copyright (2015) permission from the Royal Society of Chemistry.

*3.1. Biological Activity of Scaffolds Having Collagen Modified nHA*

The natural extracellular matrix contains fibrous proteins and growth factors, which play an important role in regulating the cells' growth and other activities. The ECM can be constituted by the combination of various constituents like arginine-glycine-asparagine, and peptides such as collagen, laminin, fibronectin, vitronectin, and HA. The adhesion, proliferation, and differentiation of diverse types of cells in ECM depend on cell-protein-hydroxyapatite interfacial interactions [65]. Several studies indicate that implants are active if they have interstitial fluids and proteins, which highlights the importance of proteins in controlling the interactions of cells with implants (scaffolds) fabricated by using protein-coated HA in the biodegradable polymer (Figure 12). In addition to this, the cellular response to the physicochemical properties of implants is also influenced by the structural organization of proteins at the surface of HA because the structural organization of the proteins at the surface of HA provides different cellular interactions.

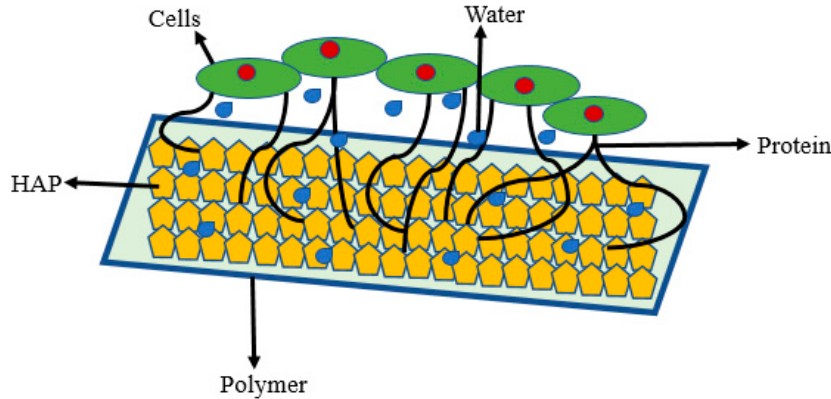

**Figure 12.** Schematic representation of HA–protein–cell interactions in ECM.

The combined use of protein, such as collagen with HA, showed an enhanced adhesion of osteoblasts; hence the bioactive polymer-based scaffolds were fabricated by using

collagen in combination with HA [66]. Since collagen is a very abundant protein in natural ECM, it is used in the fabrication of scaffolds with improved properties for tissue engineering [67,68]. Though hydroxyapatite alone is a bioactive material capable of enhancing the differentiation of osteoblasts [69–71], the synergistic effect of the combination of collagen and hydroxyapatite was found to be useful in promoting the growth of HA and bone formation [72–74]. Therefore, attempts have been made to produce various types of complexes of HA with collagen using different physical methods, including thermal HA/collagen gel assembly [75], vacuum infiltration of collagen in ceramic matrices [76], lyophilization, and critical point drying methods [77]. However, the composite scaffolds having physical bonds of collagen with HA did not show high cellular activity as compared to composite scaffolds fabricated by using chemically bonded HA with collagen. As the physical mixing of collagen and HA provides a poor bonding between collagen and HA as compared to collagen interactions with natural HA in the living system; hence, it is removed easily from the surface of HA and shows overall poor bioactivity.

Though HA has a high affinity for collagen, it becomes weaker in physiological ionic solution. Therefore, it is found to be useful to have a chemical bonding of collagen with HA to achieve high bioactivity, as observed in the interactions of bone sialo protein with collagen [56]. For example, Monkawa et al. [78] used 3-aminopropyl triethoxysilane-modified HA to enhance the chemical interactions of proteins with HA. The presence of protein and APTES at the surface of HA was confirmed by atomic force microscopy (AFM) and by recording surface $\zeta$ potential. The stability of protein/APTES/HA and protein/HA composites was evaluated by keeping these complexes in phosphate buffer and NaCl solution. The proteins immobilized on APTES/HA were found to be more stable than those immobilized on pure HA, which indicated the presence of chemical bondings between protein and HA through attached APTES. In another study, Xing et al. [79] also immobilized collagen at the surface of hydroxyapatite via covalent bonding and investigated its activity for osteoblasts. The cell proliferations and ALP activity were higher in samples having chemically bonded collagen (HA-C) than in samples having physical interactions of collagen with HA through hydrogen bonding (HA-hC). Most of the collagen adsorbed physically at the surface of HA was desorbed in the early stage of cell seeding. The reaction of APTES with hydroxyl groups of HA has produced HA surface most populated with primary amino groups that facilitated the bonding of collagen. Though APTES is classified as a toxic compound [80], its toxicity lessens on reacting its amino groups with an acid anhydride. The cytotoxicity of chemically bonded APTES at the surface of nHA was evaluated by Wang et al. [81]. In these studies, the APTES was allowed to react chemically with nHA and then neutralized with acetic anhydride or succinic anhydride before testing for cytotoxicity (Figure 13).

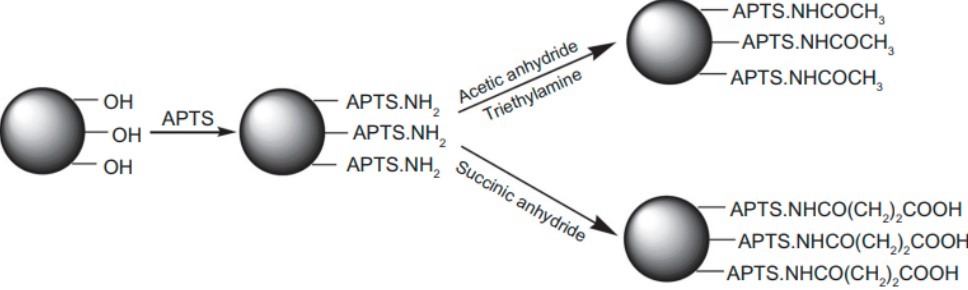

**Figure 13.** Schematic representation of reactions to modify nHA through 3-aminopropyltriethoxysilane (APTES)-mediated functionalization. Reprinted from Ref. [81] with copyright permission (2011) from Dove Medical Press Ltd.

The cytotoxicity of APTES-modified nHA was assayed by 2,5-diphenyl-2H-tetrazolium bromide (MTT) test in which yellow-colored 3-(4,5-dimethyl thiazol-2-yl)-2,5-diphenyl-tetrazolium bromide was reduced to purple-colored formazan by mitochondria reductase of

seeded live L929 cells. The produced formazan was quantitively determined by recording optical density (597 nm), which was proportional to the number of live cells; hence this MTT test was used to measure the cytotoxicity of APTES-modified nHA or other materials. The results of the MTT test did not show a statistically significant difference in cell viability between acid buffer control HA nanoparticles and APTES-modified HA nanoparticles. APTES immobilized the HA nanoparticles were able to inhibit cell viability unless they were neutralized with acetic anhydride or succinic anhydride to reduce the toxicity of APTES. Studies also indicated that the bone sialo protein in combination with triple-helical collagen, is able to promote in vitro nucleation and growth of HA [56,57]. These studies clearly indicated that orthopedic implants for bone tissue formation are more bioactive on using chemically bonded HA with collagen to have interactions of collagen with osteoblasts, as shown by collagen in natural ECM.

### 3.2. Biological Activity of Scaffolds Having Gelatin-Modified nHA

Gelatin is produced by the hydrolysis of collagen, and it has a unique sequence of amino acids. Collagen contains interconnected protein chains that produce gelatin during hydrolysis having RGD sequences of the amino acids intact, despite the hydrogen bonds breaking at the α-helix conformation of collagen (Figure 14). Gelatin is one of the most promising biopolymers and is used in the fabrication of bioactive scaffolds [44] and bioactive gels [43]. It is less expensive but has the same biocompatibility and biodegradability is similar to collagen and is composed of glycine, proline, and hydroxyproline units in its molecule.

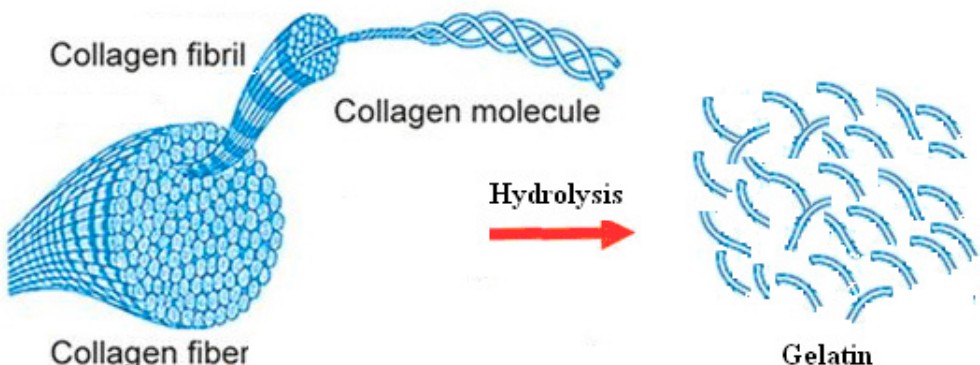

**Figure 14.** Schematic representation of the production of gelatin from collagen.

On the one hand, the brittleness in hydroxyapatite [7] limits its applications as a biomaterial. On the other hand, gelatin has more tunable mechanical property and is less osteoinductive. Thus, the combination of both gelatin and HA is complementary to produce bioactive materials for orthopedic applications [44]. The application of gelatin in the scaffolds improves cell adhesion, spreading, and cell proliferation [81–83]. The coating of gelatin on porous HA not only increases the bioactivity of HA but also increases its mechanical properties. Various investigations described the fabrication of scaffolds containing gelatin-coated HA microparticles by dropwise addition of HA gelatin dispersion in a solution of polymethylmethacrylate [84,85]. The resultant scaffolds containing gelatin–HA microparticles showed increased proliferation of human osteoblast in comparison to pure HA particles due to increased bioactivity of HA by adding gelatin at the surface of HA nanoparticles.

In another study, the gelatin and HA composites have shown enhanced biocompatibility and bioactivity for proliferation and differentiations of osteoblasts [86], which clearly indicated that the coating of gelatin was useful for increasing the bioactivity of HA particles. Gelatin is also found to be more compatible with naturally occurring polymers and can produce scaffolds with increased biomimetic properties for tissue engineering [87,88]. For example, gelatin was able to produce an injectable scaffold in combination with hyaluronan

having dose-dependent bioactivity for cell proliferation [89], but these injectable scaffolds may be more bioactive for bone tissue engineering if they are designed along with hydroxyapatite nanoparticles. The hyaluronic acid facilitates interactions with cell receptors and induces intracellular signal transduction to influence cell activities, such as proliferation, survival, movement, and differentiation.

The bioactive injectable scaffolds containing gelatin are prepared by mixing gelatin and hydroxyapatite nanoparticles with naturally occurring chitosan. These scaffolds have shown improved in vivo bone tissue formation in a subchondral bone lesion model in rabbits [90]. Similar bioactivity was also shown by photo-cross-linkable gelatin/furfurylamine scaffolds in the regeneration of subchondral bone in rabbit osteochondral defect models [91]. These investigations indicated that the chemical bonding of HA with gelatin produced scaffolds with improved stability of hydroxyapatite nanoparticles and was able to show improved bioactivity due to uniform distribution of gelatin-coated HA at the surface of scaffolds produced by using biodegradable naturally occurring polymers such as chitosan [92]. For successful implantation, the implant surface must contain at least one inorganic component, such as hydroxyapatite, and one organic component of bone, such as gelatin, to make interfacial interactions stronger to facilitate bone tissue formation [93]. Since bones are rich with collagen type I (gelatin); therefore, it is valid to believe that gelatin can act as a good candidate to act as coating materials for HA particles to enhance bone tissue formation [82,83].

To increase the bioactivity of scaffolds for bone tissue formation, the HA particles coated with gelatin and BMP-2 were mixed with biocompatible synthetic polymers such as methacrylamide and four-armed PEG methacryalmide to produce bioactive scaffolds [93]. The added BMP-2 played a significant role in increasing the interactions of nHA with polymer and reduced the aggregation of nHA that produced scaffolds with increased bioactivity of HA due to the combined effect of BMP-2 and gelatin. Scaffolds have shown a high rate of osteogenesis and were potentially useful transplants for bone tissue formation [92].

### 3.3. Bioactivity of Scaffolds Having Insulin-Modified nHA

The scaffolds for implant applications should be porous, mechanically stable, have an affinity for cell adhesion, non-toxic, biocompatible, and biodegradable. To achieve these properties in a scaffold, surface-modified porous hydroxyapatite must be dispersed in biodegradable polymers to produce scaffolds comparable with natural ECM (Figure 15).

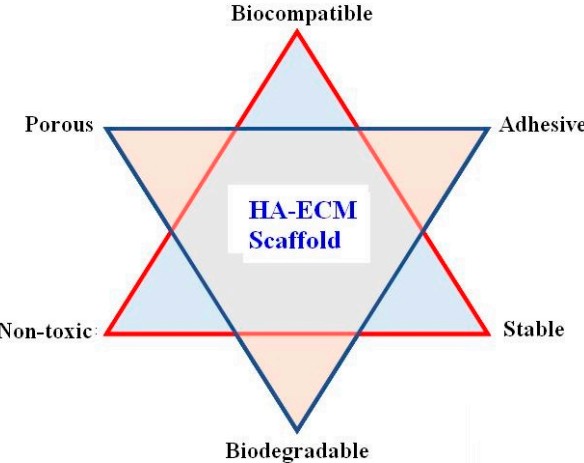

**Figure 15.** Various properties of ECM scaffolds for bone implant applications.

As the coupling of APTES for chemical bonding of a surface modifier induces toxicity in nHA [81], studies were carried out to improve the bioactivity of nHA by the chemical-bonding biomolecules instead of using APTES to improve the bioactivity of nHA [16]. In this direction, some studies were carried out to prepare scaffolds by chemical bonding

of insulin at the surface of nHA [70] using succinic acid as a spacer and completing the bonding of insulin in a two-step process. Firstly, water-soluble carbodiimide (WSC) was used to activate the succinic acid in an aqueous solution. Then, the nHA was dispersed in this solution so the succinic acid could chemically react with the nHA (Figure 16). Secondly, insulin was allowed to react chemically through its amino groups (-NH$_2$) to the carboxyl groups (-COOH) of succinic acid grafted at the surface of nHA (nHA-S) in the presence of WSC.

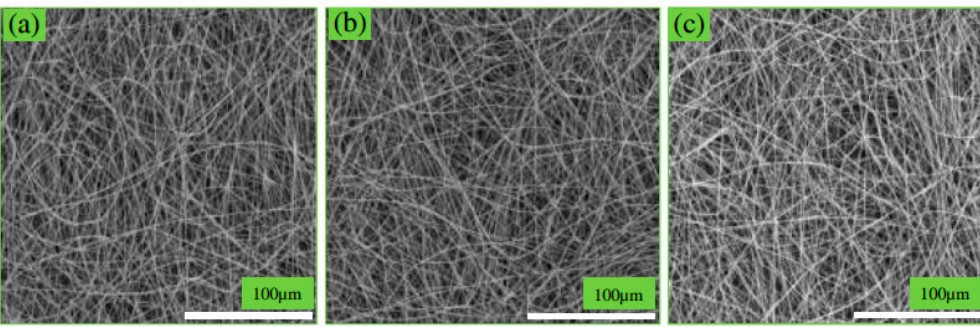

**Figure 16.** Schematic diagram depicting the grafting of insulin at the surface of nHA [70].

The nHA particles coated chemically with insulin (nHA-I) have shown high dispersion in pure organic solvents. To evaluate the bioactivity of insulin-coated nHA particles, 5–20 wt% of nHA-I particles were dispersed in a solution of PLGA in a mixed solvent of tetrahydrofuran and dimethylformamide (3:1), then electrospun as nanofiber scaffolds. After drying the nanofiber scaffolds, the scanning electron microscopic images were recorded (Figure 17). The scanning electron microscope (SEM) images of nanofiber scaffolds of pure PLGA (Figure 17a), PLGA/nHA (Figure 17b), and PLGA/nHA-I (Figure 17c) had almost the same morphology and uniform thickness irrespective of their compositions, which indicated a uniform dispersion of insulin-coated nHA particles in PLGA.

**Figure 17.** FE-SEM images of (**a**) pristine PLGA, (**b**) PLGA/nHA, and (**c**) PLGA/nHA-I nanofiber scaffolds [70].

The chemical bonding of insulin at the surface of nHA was confirmed by FT-IR spectrum. The presence of the amide I and amide II absorption bands were due to the formation of an amide bond by a reaction of the amino group of insulin with the carboxylic group of succinic acid present at the surface of succinic acid-modified nHA. The absorption band at 1740 cm$^{-1}$ in the FT-IR spectrum (Figure 18c) is attributed to the carbonyl groups

of PLGA. In addition, the absorption bands at 1650 cm$^{-1}$ and 1550 cm$^{-1}$ (Figure 18b) were due to the presence of amide I and amide-II groups, which were formed on the bonding of insulin to nHA-S [94]. The FT-IR spectrum of the PLGA/nHA-I scaffold (Figure 18d) was also having all characteristic peaks of nHA, PLGA, and insulin. These studies clearly suggested that chemically modified nHA with insulin (nHA-I) was able to produce bioactive nanofiber scaffolds having a uniform distribution of nHA at the surface of scaffolds, which enhanced the bioactivity scaffolds for osteoblasts.

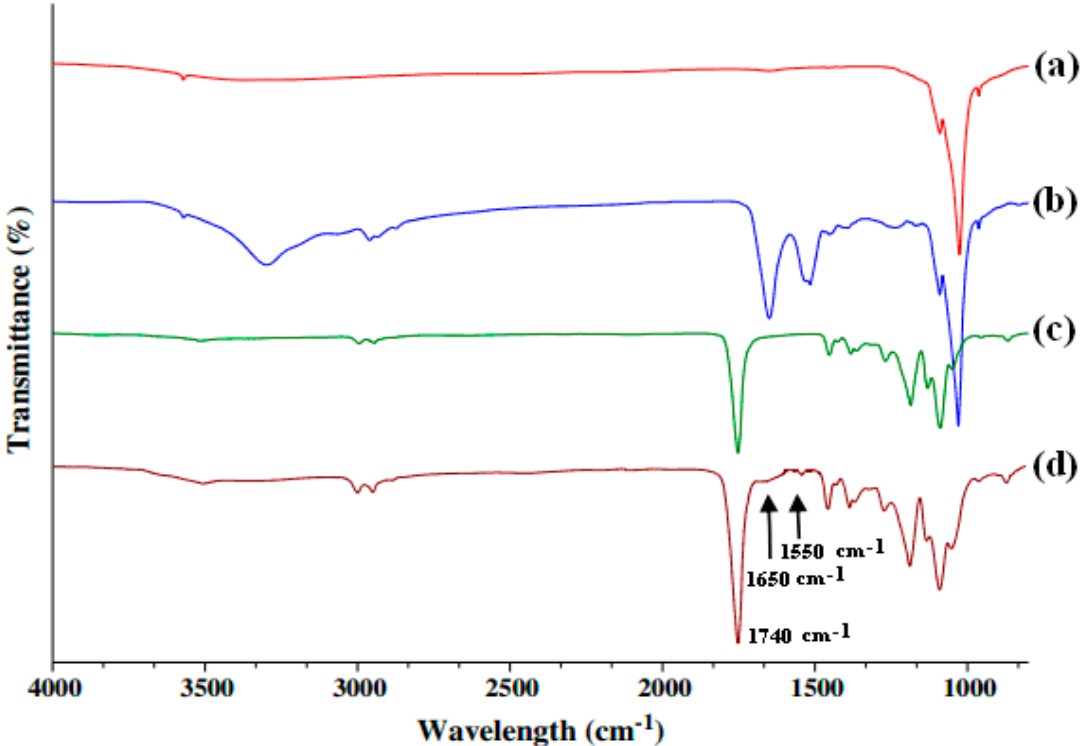

**Figure 18.** FT−IR spectra of (**a**) pristine nHA, (**b**) nHA-I, (**c**) pristine PLGA, and (**d**) PLGA/nHA-I [70].

To test the bioactivity of prepared scaffolds, the BrdU assay for cell proliferation was carried out by culturing osteoblasts on PLGA, PLGA/nHA, and PLGA/nHA-I nanofibrous scaffolds. The proliferation of osteoblasts on the nanofiber scaffolds having insulin-modified nHA, PLGA/nHA-I, was superior in comparison to the nanofiber scaffolds fabricated with pure PLGA or PLGA/nHA composites (Figure 19). These results clearly indicated that the chemical bonding of insulin with nHA played a significant role in stimulating the growth and proliferation of osteoblasts due to enhanced biocompatibility and bioactivity of nHA-I in PLGA/nHA-I nanofiber scaffolds [95,96]. The surface modification of nHA with insulin-produced nanofiber scaffolds having a high density of nHA at the outer surface of fibers increases the cell attachments and proliferations, which is beneficial to bone tissue formation. The application of insulin with nHA caused a significant increase in bioactivity of hydroxyapatite. Considering the significance of small bioactive molecules, this review also discussed the role of BMPs and peptides, which are commonly used as modifiers of bioactivity of polymer-based scaffolds [97] and are assumed as the third pillar in tissue engineering. It is worth noting that besides growth factors, the ECM proteins, adhesive peptides, hormones, cytokines, or some enzymes also provide a favorable influence on bioactivity and biocompatibility of scaffolds [49,87].

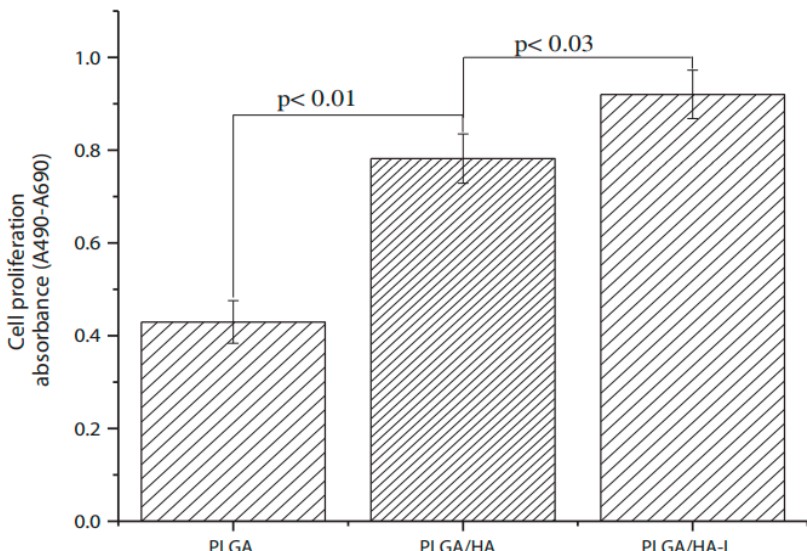

**Figure 19.** The proliferation of osteoblast cells seeded on pristine PLGA, PLGA/nHA, and PLGA/nHA-I nanofiber scaffolds for two days as determined by a BrdU assay [70].

### 3.4. Biological Activity of Scaffolds Having BMPs-Modified nHA

In vitro studies on the bioactivity of isolated mesenchyme stem cells require supports (scaffolds), bone morphogenetic proteins, and the control of biological signals (growth factors) to promote cell growth, proliferation, and differentiation to produce a particular type of tissue. The bone morphogenetic proteins belong to a group of regulatory glycoproteins that play a crucial role in controlling the growth of mesenchymal stem cells and their differentiation to chondrocytes and osteoblasts [98]. BMPs also play important roles in teeth development and differentiation of odontoblasts. Considering their osteogenic potential, the BMPs are used successfully in many therapeutic interventions, including bone defects or osteoporosis [99]. The BMP-2-derived peptides P17 and P24 were found to be promising agents in bone tissue applications [68]. The in vivo results of HA/PLGA scaffolds loaded with BMP-2 showed much better osteogenesis than PLGA scaffolds after four to eight week's post-surgery [100]. The bone bridges were more perfect and smooth than those in the PLGA scaffolds.

In line with proteins and peptides, the osteogenic and angiogenic growth factors have been used to induce and accelerate the formation of vascularized bone tissues. Among the growth factors, the BMP-2 is a well-known, strong osteogenic growth factor that promotes bone tissue regeneration [101,102]. The application of BMP-2 alone is enough to enhance bone tissue regeneration in both ectopic and orthotopic sites [103]; however, it is difficult to develop sophisticated drug-delivering bone transplants to control the delivery of growth factors and to utilize their full efficacy to enhance bone tissue regeneration [99]. The application of scaffolds loaded with growth factor-modified nHA allows for the possibility of tissue regeneration, which can overcome the issues associated with autologous and allogeneic implants. The signaling role of growth factors in the regeneration of tissue led to the development of novel strategies to deliver growth factors to various tissues. Recently, strategies to deliver growth factors to specific tissues were developed to help avoid the wastage of large amounts of growth factors administered through direct injection into the region of interest but showed limited efficacies [104].

To design scaffolds that may promote bone tissue regeneration, bioactive constructs are fabricated [21], having chemically bound BMP-2 at the surface of nHA/PLGA scaffolds. These nHA/PLGA scaffolds were prepared by the chemical bonding of L-glutamic acid at the surface of nHA (Figure 20a). The L-glutamic acid-modified nHA particles were dispersed in PLGA solution and then electrospun as nanofiber scaffolds (Figure 20b). Subsequently, BMP-2 was allowed to bind chemically through the carboxylic groups available

at the surface of nHA/PLGA scaffolds (Figure 20c). These scaffolds were designed to act as a carrier to control the release of BMP-2 to promote cell adhesion, osteoinductivity, and osteoconduction of osteoblasts to promote bone tissue formation. When analyzing the surface morphologies of the nanofiber scaffolds by their SEM images, it became clear that the BMP-2-modified scaffolds (BMP-2-g-nHA/PLGA) (Figure 21c) had almost identical surface morphologies like nHA/PLGA and PLGA scaffolds (Figure 21a,b).

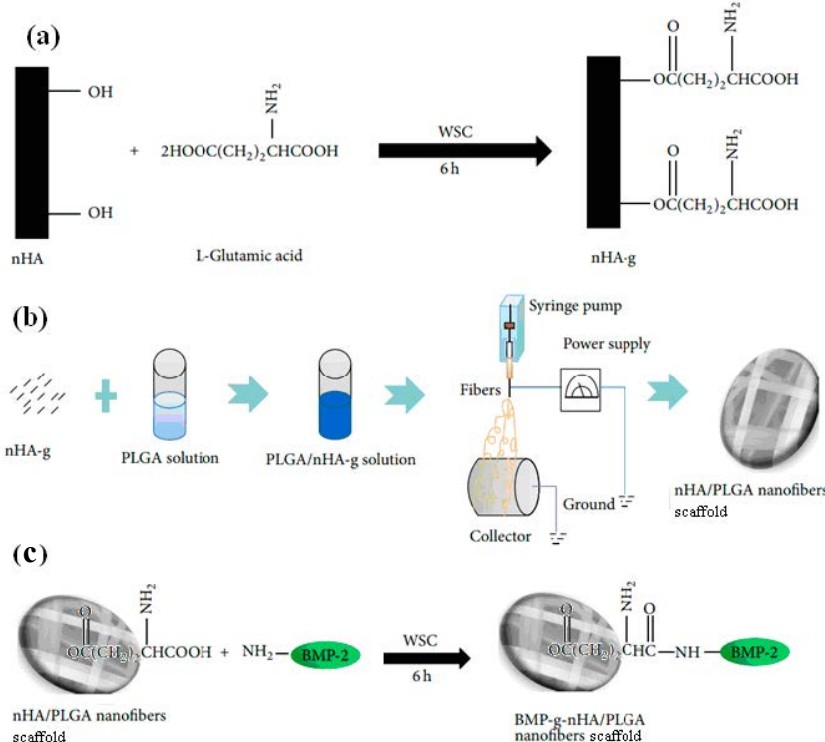

**Figure 20.** Schematic representation of BMP-2 grafting at the surface of nHA/PLGA nanofiber scaffolds. (**a**) L-glutamic acid modified nHA; (**b**) nHA filled PLGA scaffold; (**c**) BMP-2 modified PLGA scaffolds [21].

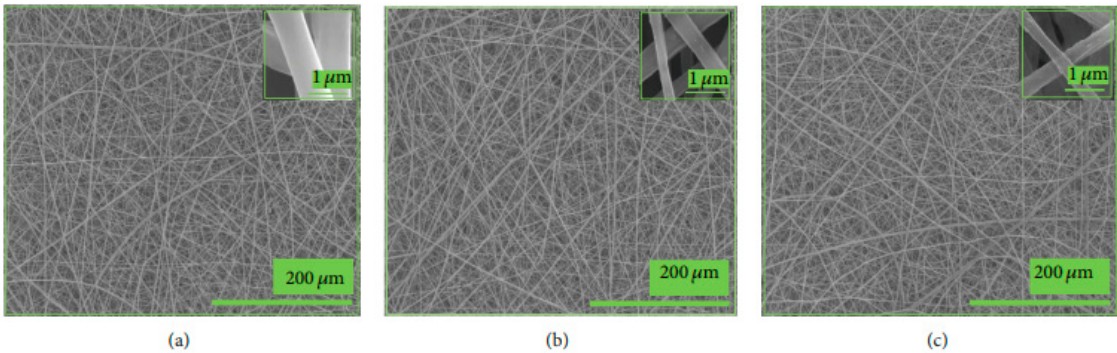

**Figure 21.** FE-SEM images of (**a**) pristine PLGA, (**b**) nHA/PLGA, and (**c**) BMP-g-nHA/PLGA nanofiber scaffolds [21].

These studies indicated that the hydroxyapatite nanorods with an average length between 40 to 100 nm rarely remained in a dispersed state in organic solvents [105] and water due to their high affinities in the formation of aggregates [106]. Therefore, to prevent the aggregation of nHA in an organic solvent, the surface of the nHA particles was modified chemically with L-glutamic acid (nHA-g), which produced a negative charge at the surface of the nHA particles, which produced repulsions amongst nHA particles and prevented nHA particles from forming their aggregates.

The transmission electron microscope (TEM) images of the nHA particles were recorded, which suggested for uniform distribution of L-glutamic acid-modified nHA particles (Figure 22b). This was further confirmed by the TEM images of the L-glutamic acid-grafted nHA particles or BMP-2-grafted nHA particles dispersed in PLGA nanofibers (Figure 22c,d). The glutamic acid-grafted HA particles showed an increased affinity with hydrophobic PLGA and were thus well-dispersed in the PLGA without forming any aggregation. To confirm the bonding of BMP-2 at the surface of nHA/PLGA nanofiber scaffolds, the fluorescein isothiocyanate (FITC) labeled BMP-2 was attached chemically at the surface of the nHA/PLGA nanofiber scaffolds, and fluorescence images were recorded using a confocal fluorescence microscope. The appearance of the green image (Figure 23) clearly confirmed that BMP-2 was attached with nHA particles and distributed uniformly at the surface of the nHA/PLGA nanofiber scaffolds.

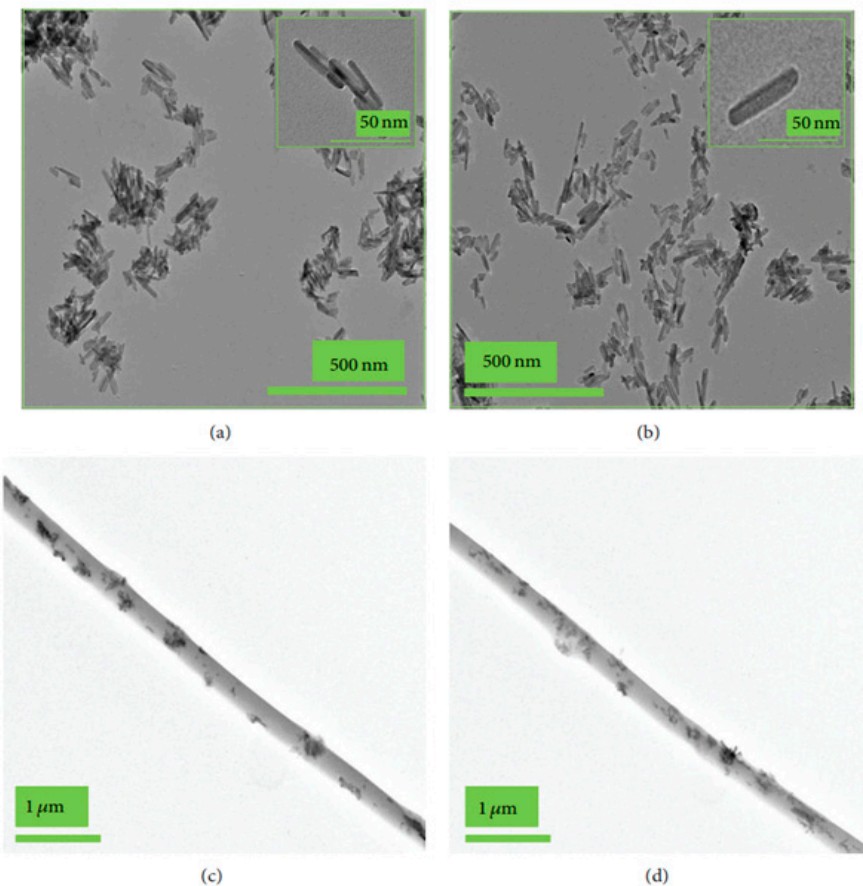

**Figure 22.** TEM images of (**a**) pristine nHA, (**b**) nHA-g, (**c**) nHA/PLGA, and (**d**) BMP-g-nHA/PLGA nanofiber scaffolds. The subsets on (**a**,**b**) are to show highly magnified TEM images of pristine nHA and nHA-g, respectively [21].

The scaffolds having nHA particles with chemically bonded BMP-2 at their surfaces were potentially more bioactive in bone formation, as the bonded BMP-2 increased the synthesis of osteocalcin/mineralization due to the added osteogenic properties by BMP-2 in the BMP-g-nHA/PLGA scaffolds. The osteogenic properties of BMP-g-nHA/PLGA scaffolds were determined by analyzing the deposited calcium [21] by the alizarin Red staining method after seeding osteoblasts at clean PLGA, nHA/PLGA, and BMP-g-nHA/PLGA nanofiber scaffolds for 15 days. The results shown in Figure 24 suggest that the nHA/PLGA nanofiber scaffolds (Figure 24b) had a better deposition of calcium than pristine PLGA nanofiber scaffolds (Figure 24a). However, the BMP-g-nHA/PLGA scaffolds were found to be highly osteogenic and showed the highest deposition of calcium (Figure 24c).

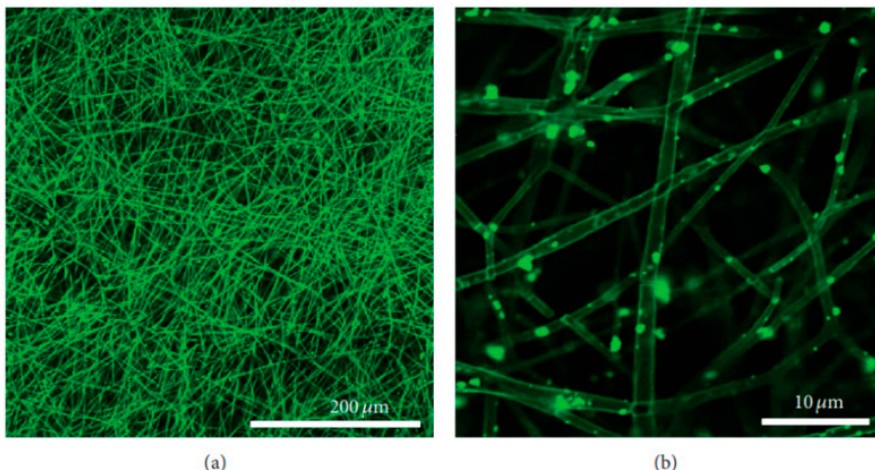

**Figure 23.** Confocal fluorescence images of FITC-labeled BMP-g-nHA/PLGA hybrid nanofiber scaffolds at low (**a**) and high (**b**) magnifications [21].

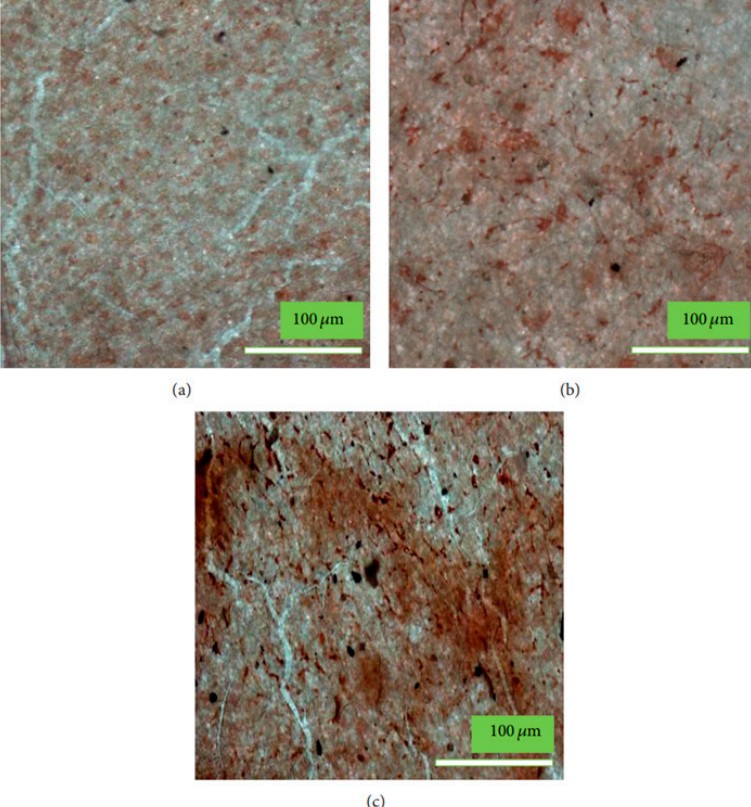

**Figure 24.** Alizarin Red staining of osteoblasts seeded for 15 days on (**a**) PLGA, (**b**) nHA/PLGA, and (**c**) BMP-g-nHA/PLGA nanofiber scaffolds [21].

These results clearly suggest that chemically bonded growth factors (biomolecule), such as BMP-2 or insulin at the surface of nHA or scaffolds, were quite active in producing scaffolds with enhanced differentiation of osteoblasts for bone tissue formation [21,70]. When the nHA particles bonded chemically with the bioactive molecule (Figure 25b) and mixed with biodegradable polymers, there were then able to disperse homogenously in polymers matrices. The scaffolds prepared from these dispersions are more bioactive than scaffolds (Figure 25a), having nHA without a biomolecule, as confirmed by other studies [21].

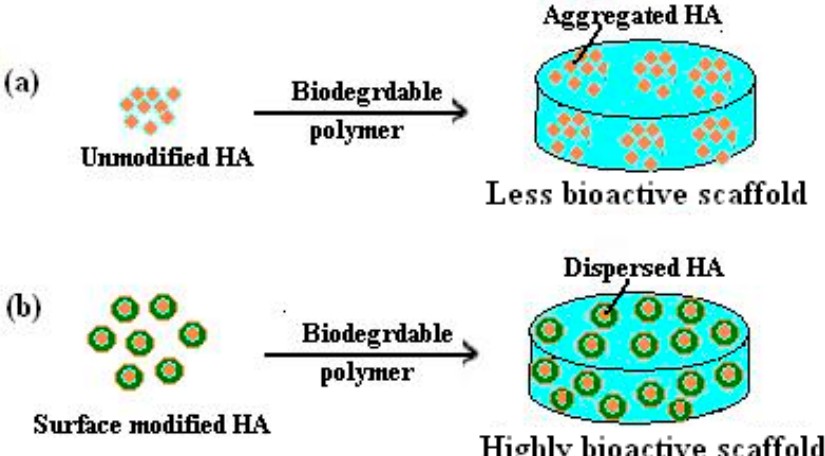

**Figure 25.** Schematic diagram depicting the effect of surface modification of nHA by biomolecules on the bioactivity of scaffolds. (**a**) scaffold with nHA; (**b**) scaffold with modified nHA.

### 3.5. Bioactivity of Scaffolds Having nHA Modified with Miscellaneous Molecules

The PLA is grafted on nHA to improve the dispersion of nHA in PLGA and to increase the overall bioactivity of the resultant scaffolds [100]. The dispersion and bioactivity of PLA-modified nHA have shown significant improvement by adding a small amount BMP-2 in PLGA scaffolds due to the cell-binding ability of added proteins at the surface of scaffolds. The grafting of PLA oligomers at the surface of nHA has improved the phase compatibility of nHA with PLGA [92]. To improve the phase compatibility of nHA in PLGA, Zhang et al. [107] used polylactic acid grafted nHA in PLGA composite scaffolds that showed improved mineralization and osteogenic properties of the scaffolds. Similarly, bioresorbable fiber membranes for guided bone regeneration were prepared using PLA-modified nHA in PLGA [108]. These studies indicated that the nHA particles were dispersed more effectively in polymer scaffolds by using a phase compatibilizer, and increased stability was due to the chemical bonding of small molecules at the surface of nHA.

In other studies, the hydrophilic poly (D, L-lactide) was grafted chemically at the surface of nHA by ring-opening polymerization of D, L-lactide using hydroxyl groups of nHA as initiating centers [109–112]. In these studies, the chemical bonding between nHA particles and poly (D, L-lactide) was confirmed by FT-IR and nuclear magnetic resonance (NMR) methods, and D, L-lactide grafted nHA particles have shown improved biocompatibility and osteoconductivity. Though this method of D, L-lactide grafting at the surface of nHA is easy. However, with this method, it was difficult to control the reaction parameters to limit the molecular weight of grafted poly (D, L-lactide) at the surface of nHA.

For the use of poly (L-lactic acid) (PLLA) as a scaffold, a wide range of scaffolds for bone tissue engineering can be created by binding poly(γ-benzyl-L-glutamate) to the surface of the nHA particles and dispersing them in PLLA [113]. High bioactive poly-caprolactone (PCL) scaffolds for bone tissue engineering were produced by dispersing dopamine-modified nHA in PCL [114]. In these studies, dopamine is used as an effective bioadhesive and to distribute nHA particles uniformly at the surface of PCL scaffolds; hence resultant scaffolds were more bioactive in nucleation and growth of nHA for bone formation.

To compatibilized, the nHA with the polymer and the property of the interface between the polymer and nHA were also controlled by the chemical modification of nHA by stearic acid (Sa). In this regard, Li et al. [20] added stearic acid to the surface of the nHA particles. For this purpose, stearic acid was dissolved in an organic solvent and refluxed for several hours before coating on porous nHA particles. The coating of Sa on porous nHA particles was confirmed by comparing its FT-IR spectrum with the FT-IR spectrum of neat

nHA particles. The coating of Sa on nHA particles was found to be unstable due to the physical adsorption of Sa on porous nHA. To achieve a stable coating of Sa on the porous nHA particles, a chemical bonding of stearic acid with nHA particles is desired, which is a prospective area for further studies. The chemical bonding of physiologically active molecules at the surface of the nHA particles increases the dispersion of nHA polymers, as well as improves the physiological activities of prepared scaffolds. In addition, polyesters and various polysaccharides such as cellulose, chitosan, and agarose were also used to compatibilize nHA particles with the organic phase to prepare bioactive scaffolds for bone tissue formation [115]. Chitosan is a biodegradable polymer and antibacterial in nature; hence, it is also found to be a useful biomaterial for the fabrication of scaffolds for tissue engineering.

### 3.6. Effect of Irradiation on Antibacterials and Bioactivity of Hydroxyapatite Particles

Hydroxyapatite, in combination with $TiO_2$/yttria-stabilized zirconia/alumina /nanodiamond/magnesium, or a natural fiber, has been used to produce HA composites to enhance bioactivity and mechanical properties of scaffolds for bone implant applications [116]. Similarly, the $ZnO/SiO_2$ or $Ag_2O$-doped plasma-sprayed hydroxyapatite coating was also prepared with improved mechanical and antibacterial activities for orthopedic and dental applications [117–119]. These nHA coatings have shown increased antibacterial properties and high biocompatibility indispensable for biomedical applications. A multifunctional composite of photocatalyst ($TiO_2$) and plasma-sprayed nHA-amino acid coating was also prepared, which showed improved antibacterial activities and cell adhesion on irradiating with visible light [120,121] to avoid the damaging effects of high energy radiations. However, such radiation-induced production of nHA coatings is found to be associated with some limitations owing to the changes that occurred in nHA on heating during plasma spraying. The excessive heat of plasma leads to produce crystalline nHA, and nHA also transformed to a soluble phase on thermal decomposition; hence, further studies are still required though some studies are carried out to improve bone cell adhesion and antibacterial activities in plasma-sprayed nHA composite coatings by using suspension plasma spray (SPS) coating technique.

The SPS method led to the production of crystalline nHA biomaterial for coating without having the adverse effect of heating on nHA [122]. By controlling the SPS parameters, the bioactive but less crystalline nHA coating was developed. The antibacterial activity of suspension plasma sprayed hydroxyapatite was evaluated by determining the number of colony-forming units (CFU) [123]. The antibacterial activity of suspension plasma-sprayed hydroxyapatite was evaluated by using *E. coli*, and cell viability was evaluated by live/dead assaying of cells at the surfaces of these nHA coatings after light irradiation. The adhesion and viability features of *E. coli* were evaluated by using a fluorescence microscope. The nHA/gray titania coating has shown a nonhomogeneous surface along with microporosity. It contained the magneli phase (oxygen-deficient titanium oxides) produced after irradiation, but it did not show any adverse effect on the viability of osteoblasts, as was clear in these studies [120]. Thus, the variation in bioactivity and antibacterial properties of nHA-$TiO_2$ composites on plasma irradiation is considered due to phase transition in nHA particles and due to the formation of reactive oxygen species (ROS), such as OH and superoxide radicals, by the attached $TiO_2$ upon irradiation to light. The antibacterial nHA particle coating is also found useful in dental implants to prevent bacterial infection in an artificial dental root, which causes difficulties in the treatment of the surrounding region of dental implants. The antibacterial effect in nHA coating can be produced by the combination of various metals such as Ag and Mg with HA or by a combination of nHA with light-sensitive photocatalysts such as $TiO_2$. When nHA-based composite coatings are used for dental implants, then visible light-sensitive photocatalysts are generally preferred due to safety reasons [121,124].

The variation in the bioactivity of nHA coating on irradiation is theoretically explained on the basis of alternation in deepest structures on electron excitation in nHA and due to

the influence of various types of defects such as vacancies, internodes, as well as due to the presence of protons and hydroxyl groups on the surface of the nHA particles [25]. The photoelectron excitation of nHA is able to produce defects maximum up to a level of 6 eV from the surface, but theoretical calculations indicated that the synchrotron technique of irradiation of nHA provides opportunities to study the electron excitation effect on the surface properties of nHA by exciting the electron from the deepest level of 35 eV. The structures so formed at a depth of 35 eV by synchrotron irradiation technique are able to cause a significant variation in bioactivity due to variation in excess charge density at the surface of the nHA particles. Theoretical calculations were carried out for the shift in densities of energy states (DOS) to relate the effect of lattice's charge variation on the bioactivity of the nHA particles for living cells [125,126]. These theoretical calculations for the variations in surface properties of the nHA particles on irradiation are useful to relate the bioactivity of nHA with H and OH vacancies created on excitation of electrons on irradiation from the different levels of lattice cells of nHA. Thus, nHA coatings of desired cell activities could be designed by irradiation of the nHA particles with suitable radiation.

## 4. Conclusions

Since hydroxyapatite is chemically identical to the hydroxyapatite naturally present in bones; hence, it is widely used as a coating material for metal implants for their fixation and regeneration of bones damaged by various types of bone diseases. For the coating of the nHA particles on the surfaces of the bone implants, various methods are used, such as sputtering [127], ion plating, and laser irradiation [128]. However, these physical methods for the deposition of nHA particles on metal surfaces mostly end up with delamination, aggregation, and destruction of the interface. Therefore, a second surgery is required for the fixation of bone implants.

To overcome these drawbacks of bone implants, the application of the nHA particles for an implant coating modified with bi-functional biomolecules or small organic molecules [129] is proved to be potentially useful. This technique can compatibilize the implants with bone tissues and the environment to promote the formation of nHA-like natural ECM. The chemically modified nHA can reduce the possibility of interfacial delamination, aggregation, and interfacial failures of metallic bone implants. Besides these advantages of modified nHA, the chemically modified nHA with bi-functional small molecules can show significantly high dispersibility in aqueous or organic solutions of polymers, which are used for the fabrication of bioactive implants (scaffolds) for bone tissue formation. Since surface-modified nHA particles are well mixed with biodegradable polymers such as polyesters, chitosan, collagen, and biocompatible and bioactive materials are produced easily.

To add functional groups at the surface of the nHA particles, inorganic molecules such as APTES are a common coupling agent that is bonded chemically with abundant hydroxyl groups present at the surface of ceramics and create reactive groups for bonding at the surface of metal implants [130]. In addition to APTES, various other molecules such as sugars, amino acids, proteins, peptides, drugs, etc., were found to be useful for chemical modification of nHA as these molecules are non-toxic and able to add various types of surface functionality such as amino ($-NH_2$), carboxylic acid (-COOH), and hydroxyl (OH), which facilitate the interactions of nHA with implants and bone cell for bone tissue formation. As the toxicity of APTES is controversial in applications related to the implant surface, the applications involving biomolecules seem to be a preferred substitute for APTES.

The surface modification of HA nanoparticles is more useful as it prevents the agglomeration of nHA [131] besides causing enhanced interactions with implants and bone cells. The porosity in nHA particles is also found to be useful in increasing the interfacial area, cell seeding, and loading of drugs when scaffolds are used for the delivery of the drugs at the site of bone diseases. Thus, there is also a need to develop porous nHA for designing drug-loaded scaffolds for various types of bone diseases to improve the efficacies

of commonly used drugs [132], which otherwise were shown to have low efficacy on administration by routine delivery via injections. There is a need to study the biological effect of nanopores in nHA having sizes in tens of nanometers range as similar to those as found in natural bones. This will facilitate the loading of various types of growth factors or drugs at the surface of modified nHA as used in polymer-based scaffolds for bone tissue formation. Recent investigations [21,70] indicated that the bioactivity of the nHA particles can easily be improved by direct chemical bonding of natural organic acids or amino acids at the surface of the nHA particles. These possibilities for direct chemical bonding with nHA particles or polymer scaffolds were attempted with a few drugs such as pamidronic acid [20], and cell growth factors such as BMP-2 [21] to fabricate the bioactive and drug-releasing scaffolds for bone tissue engineering. However, these must be studied further with other drugs and growth factors.

The surface bioactivity of the nHA particles for cell adhesion and antibacterial activity could be controlled by designing nHA particles that would produce useful surface morphologies by irradiating nHA particles with visible light by causing structural changes in the lattices of the nHA particles. The presence of a photocatalyst is able to create ROS to add antibacterial activity to the nHA particles; hence, a combination of different photocatalysts with different band structures with nHA particles would provide an opportunity to control radiation-induced bioactivity of the nHA particles. The studies reviewed in this article have shown tremendous opportunities and future prospects for further studies to develop physiologically active substances to modify the surface of nHA through chemical bonding to promote bone regeneration and to develop therapeutic bone implants for the treatment of various bone diseases, such as osteoporosis.

**Author Contributions:** Conceptualization, S.K. and A.H.; methodology, K.C.G. and H.K.; validation, S.K.; investigation, A.H.; writing—original draft preparation, I.K.; writing—review and editing, K.C.G.; project administration, H.K.; funding acquisition, H.K. and I.K. All authors have read and agreed to the published version of the manuscript.

**Funding:** This work was supported by the 2021 Yeungnam University research grant and the Technology Development Program (S3005767) supported by the Ministry of SMEs and Startups (MSS, Korea).

**Institutional Review Board Statement:** Not applicable.

**Informed Consent Statement:** Not applicable.

**Data Availability Statement:** Not applicable.

**Conflicts of Interest:** The authors declare no conflict of interest. The funders had no role in the design of the study, such as the collection, analyses, or interpretation of data, writing of the manuscript, or the decision to publish the results.

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
