# Peer review of "Chemical Bonding of Biomolecules to the Surface of Nano-Hydroxyapatite to Enhance Its Bioactivity"

_coatings, doi:10.3390/coatings12070999_

Round 1
Reviewer 1 Report
This version does not look worthy and cannot be recommended for publication in this form and needs some revision.
1. It is recommended that in the introduction, in the form of a short paragraph, brief information about the chemistry and structure of HAP should be given. Surprisingly, HAP chemical formula is not given.
2. Discussing polymer-based implants, it is important to give a clearer description of fundamentals, how inorganic material interacts with polymers (See, recent very detailed analysis in the case of TiO2:
Tsebriienko, T.; Popov, A.I. Effect of Poly(Titanium Oxide) on the Viscoelastic and Thermophysical Properties of Interpenetrating Polymer Networks. Crystals 2021, 11, 794. https://doi.org/10.3390/cryst11070794
3. A significant shortcoming of the review is the absence of generalizing tables, which always attract a large number of readers. I would like to see in the new version of the manuscript less text and more generalizations
4. Some figures may be omitted because they do not contain sufficiently striking results.
For example, 16, 21, 23.
Author Response
Reply to questions of learned referee I
General comments: This version does not look worthy and cannot be recommended for publication in this form and needs some revision.
Reply: Authors are thankful to referee for constructive comments and opportunities to revise this manuscript.
Q1-It is recommended that in the introduction, in the form of a short paragraph, brief information about the chemistry and structure of HAP should be given. Surprisingly, HAP chemical formula is not given.
Reply: As suggested by referee, a brief detail about synthesis of HAP along with structural characteristics in the form of FT-IR and XRD spectra (Figure 1a & b) are included at Pages 2-3.
Q2- Discussing polymer-based implants, it is important to give a clearer description of fundamentals, how inorganic material interacts with polymers (See, recent very detailed analysis in the case of TiO2:
Reply: The role of addition of nHA and types of polymer-inorganic framework
interactions on properties of scaffolds are discussed highlighting the type of
interfacial interactions from chemical point of views. The correction is added at
Page 4. The influence of addition of nHA on elastic modulus in also explained as
suggested by learned referee at Page 4.
Q3- A significant shortcoming of the review is the absence of generalizing tables, which always attract a large number of readers. I would like to see in the new version of the manuscript less text and more generalizations.
Reply: Authors agree with the suggestions of referee to provide information on tabular form but that is usually possible when there is enough data from various
workers on proposed concepts but very few studies are available, hence efforts are made to provide concept both in the form of text and figures (such as figure -2, Figure 12, Figure 15, Figure 25) as well as in some tabular forms (Table 1 and Table 2). We regard the idea of learned referee but due to these limitations, the present manuscript can’t be changed largely in tabular form as desired. We hope learned referee agrees and satisfied with our reply for his suggestion.Q4- Some figures may be omitted because they do not contain sufficiently strikingResults. For example, 16, 21, 23.
Reply: These figures are important to describe the distribution of HA at the surface of scaffolds and to authenticate the role of HA in controlling the bioactivity of the scaffolds. Further they are authors own work to provide biological test support on distribution of nHA and on bioactivity of the scaffolds on adding nHA in polymer scaffolds, hence we hope learned referee agree with our reply and allows to keep them as such in the interest of the readers. Figures are also substitute of tabular presentation of data.
Finally, we hope our replies satisfy the learned referee.

Reviewer 2 Report
Dear Authors,
I don't have any questions. The article is made at a high level, the description of the experiments, the studies carried out are of scientific interest.
Author Response
Reply to questions of learned referee II
Comments: I don't have any questions. The article is made at a high level, the description of the experiments; the studies carried out are of scientific interest.
Reply: Authors are thankful to referee for constructive and critical evaluation of the manuscript.

Reviewer 3 Report
This manuscript presents a review about the chemical modification of the surface of nano-hydroxyapatite by bonding biomolecules for enhancing its bioactivity. The review is well done, structured and referenced, but some aspects, mainly formal, must be corrected to uniformise the notation of the different sections that probably have been written by different authors. Those aspects are important in the case of a review and must be corrected before its publication.
1) A lot of acronyms are used along the manuscript, and the mostly are defined, but there are some acronyms not defined and other defined in each of the sections of the manuscript. Therefore, on one hand all the acronyms must be defined the first time that appear to facility their comprehension by all the readers. On the other hand, those acronyms more known can be defined once and avoid the excessive repetition of its meaning. One possibility would be to include a glossary of terminology. For example, in the introduction section, the first time that appear BMP-2 growth factor (line 68) or RGDs sequence (line 74), are not defined. In other sections PVA (line 204), APTES (line 308), PEG (line 309), THF and DMF (lines 611-12) must be also defined. The same indication applies for the techniques mentioned that must be defined the first time that appear (FTIR line 249 and figure 5, HETCOR spectra in line 435, NMR in line 456, and SEM and TEM in figures 20 and 21 respectively.
2) The pamidronic acid is abbreviated in three different ways along the manuscript: P, Pa, PDA. The authors must use the same notation along the manuscript. See lines 239, 299 and 312.
3) Figure 7 is not clear, because the notation of figure is not indicated either in the text and the captions of the figure.
4) What is the meaning of CaP in line 336?
5) Authors should explain briefly what the ALP activity (line 506) and MIT test (line 529) are.
6) Some other formal aspects as interline space between paragraphs and margin justification in the different paragraphs must be revised and unified in all the text.
Author Response
Replies to questions of learned referee III
General comments: This manuscript presents a review about the chemical modification of the surface of nano-hydroxyapatite by bonding biomolecules for enhancing its bioactivity. The review is well done, structured and referenced, but some aspects, mainly formal, must be corrected to uniformise the notation of the different sections that probably have been written by different authors. Those aspects are important in the case of a review and must be corrected before its publication.
Reply: Authors are thankful to learned referee for appreciation and constructive comments on our manuscript. We considered the suggested corrections for notations and other points and manuscript is revised to the satisfaction of learned referee.
Q1- A lot of acronyms are used along the manuscript, and the mostly are defined, but there are some acronyms not defined and other defined in each of the sections of the manuscript. Therefore, on one hand all the acronyms must be defined the first time that appear to facility their comprehension by all the readers. On the other hand, those acronyms more known can be defined once and avoid the excessive repetition of its meaning. One possibility would be to include a glossary of terminology. For example, in the introduction section, the first time that appear BMP-2 growth factor (line 68) or RGDs sequence (line 74), are not defined. In other sections PVA (line 204), APTES (line 308), PEG (line 309), THF and DMF (lines 611-12) must be also defined. The same indication applies for the techniques mentioned that must be defined the first time that appear (FTIR line 249 and figure 5, HETCOR spectra in line 435, NMR in line 456, and SEM and TEM in figures 20 and 21 respectively.
Reply: All the acronyms are now defined and made them uniform without repetition of full names once they are defined except in occasional situation where full form is required.
Q2- The pamidronic acid is abbreviated in three different ways along the manuscript: P, Pa, PDA. The authors must use the same notation along the manuscript. See lines 239, 29 and 312.
Reply: The acronym for pamidromic acid is now unified as Pa as pointed out by referee.
Q3- Figure 7 is not clear, because the notation of figure is not indicated either in the text and the captions of the figure.
Reply: As pointed out by referee figure 7 (now figure 8) is made more clear. It is basically a scheme for the addition of pamidronic acid (Pa).
Q4- What is the meaning of CaP in line 336?
Reply: The CaP is a general acronym for a particular form of calcium phosphate (calcium orthophosphate), which now defined at Page 11.
Q5- Authors should explain briefly what the ALP activity (line 506) and MIT test (line 529) are.
Reply: Thanks for useful comments. Details of ALP and its importance is defined at Page 3 and at Page17 for MTT assay to make these protocol useful to readers.
Q6-Some other formal aspects as interline space between paragraphs and margin justification in the different paragraphs must be revised and unified in all the text.
Reply: Manuscript is revised to unified spacing, paragraphs, and margins as pointed out by learned referees.
Finally, authors are thankful to learned referee for useful comments and opportunities to revise the manuscript worth consideration for publications in coating journal.

Reviewer 4 Report
The review 'Chemical Bonding of Biomolecules to the Surface of Nano-hydroxyapatite to Enhance its Bioactivity' is devoted to surface modification of HAp for the purpose to increase its biocompatibility and bioactivity, as well as to improve its antibacterial properties. Quality of the work meet high level of the Coatings journal, and I recommend to accept this manuscript after minor revision according to recommendations below.
I recommend to add brief description of the crystal structure of HAp and its analogs into the Introduction, taking into account non-equivalence of the crystal surfaces [for example, see Phys. Rev. B 70 (2004) 155104, https://doi.org/10.1103/PhysRevB.70.155104; Japan. Dent. Sci. Rev. 51 (2015) 85; https://doi.org/10.1016/j.jdsr.2015.03.004]. The Fig. 9 seems not appropriate in p. 11, fundamentals of the chemical bonding of HAp with 'small molecules' should be introduced earlier.
I also recommend to emphasize the problem of the compatibility of HAp with biodegradable polymers:
[Int. J. Mol. Sci. 22 (2021) 7690, https://doi.org/ 10.3390/ijms22147690, and references therein].
And to refer on:
[Chem. Rev. 108 (2008) 4754, https://doi.org/10.1021/cr8004422]
[Biomater. Res. 23 (2019) 4, https://doi.org/10.1186/s40824-018-0149-3]
[RSC Adv. 8 (2018) 2015, https://doi.org/10.1039/c7ra11278e]
[Mater. Today 19 (2016) 69, https://doi.org/10.1016/j.mattod.2015.10.008]
[Emergent Mater. 3 (2020) 521, https://doi.org/10.1007/s42247-019-00063-3].
[J. Biomed. Mater. Res. B Appl. Biomater. 106 (2017) 2046, https://doi.org/10.1002/jbm.b.33950].
[Appl. Sci. 10 (2020) 3483, https://doi.org/10.3390/app10103483].
[Coll. Surf, B: Biointerfaces, 173 (2019) 171, https://doi.org/10.1016/j.colsurfb.2018.09.074].
Author Response
Replies to questions of learned referee IV
General comments: The review 'Chemical Bonding of Biomolecules to the Surface of Nano-hydroxyapatite to Enhance its Bioactivity' is devoted to surface modification of HAp for the purpose to increase its biocompatibility and bioactivity, as well as to improve its antibacterial properties. Quality of the work meet high level of the Coatings journal, and I recommend to accept this manuscript after minor revision according to recommendations below.
Reply: Authors are thankful to learned referee for the positive and constructed comments. As suggested, a brief description on crystal structure and role of surface properties of HAP in adsorption of biomolecules is indicated considering ions substitution and modification with organic molecules in introduction at Page 3 considering the suggested references by referee.
Q-I recommend to add brief description of the crystal structure of HAp and its analogs into the Introduction, taking into account non-equivalence of the crystal surfaces [for example, see Phys. Rev. B 70 (2004) 155104, https://doi.org/10.1103/PhysRevB.70.155104; Japan. Dent. Sci. Rev. 51 (2015) 85; https://doi.org/10.1016/j.jdsr.2015.03.004].
Reply: A brief description on structure of HAP and its surface anisotropy is given in introduction at Page 3 and suggested references also added (Phys. Rev. B 70 (2004) 155104, https://doi.org/10.1103/PhysRevB.70.155104; Japan. Dent. Sci. Rev. 51 (2015) 85; https://doi.org/10.1016/j.jdsr.2015.03.004). These references are cited as Reference 9 and 10.
Q-2-The Fig. 9 seems not appropriate in p. 11, fundamentals of the chemical bonding of HAp with 'small molecules' should be introduced earlier.
Reply: I hope learned referee will agree with Figure 9 (Now Figure 10) in support of hexagonal structures of HA and about anisotropic nature of HA and to support the area of different planes responsible for adsorption and directional growth of the HA. The importance of adsorption of small is supported with relevant reference [45]. The basis for chemical bonding of HAP is explained on the basis of surface modification of HAP with organic molecules which are able to form chemical bonding with small molecules such as with insulin Page 21. To explain the chemical bonding of small molecules such as pandronic acid, authors have described the surface modification with APTES other molecules such as steric acid (Page 3, Ref-20). As desired, the surface modification for chemical is given introduction.
Q-3 I also recommend to emphasize the problem of the compatibility of HAp with biodegradable polymers:
Reply: Although the problem of compatibility of HA and nanoparticles in biodegradable polymers was already highlighted and to overcome this problem, the importance of surface modification of HA was also summarise in Fig 25. The discussion is further authenticated with suggested references such as the role of copolymer with phosphate unit in compatibilization of HA, which is supported with suggested reference. Following references as suggested by learned are added to support the aspect of compatibilization of HA in biodegradable polymers.
Int. J. Mol. Sci. 22 (2021) 7690 and references therein (Added at Page 4 with Reference number 22)
Chem. Rev. 108 (2008) 4754, (Added at Page 16 with reference number 66)
Biomater. Res. 23 (2019) 4,] (Added at Page 17, with reference number 43)
RSC Adv. 8 (2018) 2015, (Added at Page 1, with reference number 4)
Mater. Today 19 (2016) 69, (Added at Page 1, with reference number 5)
Emergent Mater. 3 (2020) 52 (Added at Page 1, with reference number 3)
- Biomed. Mater. Res. BAppl. Biomater.106 (2017) 2046, (Added at Page 4 with Reference number 17).
Appl. Sci. 10 (2020) 3483, (Added at Page 13, with reference number 50).
Coll. Surf, B: Biointerfaces, 173 (2019) 171(Added at Page 6 with reference number 30).
Finally, we thanks learned to suggest useful reference that helped in improving the discussion on compatibilization of HA in biodegradable polymers. We hope it satisfy the learned referee.

Round 2
Reviewer 1 Report
Please check references 23, 46, 66 !!!
There are given names instead of surnames.
Correct version:
23. Tsebriienko, T.; Popov, A.I. Effect of Poly(Titanium Oxide) on the Viscoelastic and Thermophysical Properties of Interpenetrating Polymer Networks. Crystals 2021, 11, 794. https://doi.org/10.3390/cryst11070794
46. Florea, D.A.; Chircov, C.; Grumezescu, A.M. Hydroxyapatite Particles—Directing the Cellular Activity in Bone Regeneration Processes: An Up-To-Date Review. Appl. Sci. 2020, 10, 3483.
66. Palmer, L.C.; Newcomb, C.J.; Kaltz, S.R.; Spoerke, E.D.; Stupp, S.I. Biomimetic systems for hydroxyapatite mineralization inspired by bone and enamel. Chem. Rev. 2008, 108, 4754–4783.
Author Response
Thank you very much for your kind review. We corrected refs 23, 46, 66 with incorrect author notation exactly as you pointed out. The corrected refs are marked in red color in the text. Thank you again.
This manuscript is a resubmission of an earlier submission. The following is a list of the peer review reports and author responses from that submission.
Round 1
Reviewer 1 Report
The authors review recent research in hydroxyapatite, mainly from the point of enhancing its nanoparticle dispersibility and interfacial binding. However, in my personal opinion, I think that perhaps the depth and breadth of literature research in the current article is not very good, and no meaningful discussion or review is given.
Reviewer 2 Report
This paper is very narrow on its scope and does not provide any substantial benefit if published.
The bulk of the paper is a summary of 5 papers (references: 15, 16, 17, 20 and 22). The other papers just get a short mention in the paper. The subtopics are also not clearly thought out. All this are major negative points.
If the authors want to resubmit this paper, the comments above need to be addressed.
Reviewer 3 Report
This is a rather interesting topic, which, of course, is needed in the development and promotion, the results that are reviewed are interesting and can be accepted for publication after a more detailed disclosure of some ambiguities and uncertainties.
- Line 35. Review [1] is quite old, here it is necessary to add some more recent and relevant works. See for example some of them in
https://www.mdpi.com/search?q=Hydroxyapatite+
- the authors forgot to discuss in this review the tremendous importance of radiation aging. Please discuss if there is radiation aging in considered HAP materials and how it is important? It is known that intense high energy UV, X-ray or electron may produce some aging of HAP. See: Bystrova, A.V., Dekhtyar, Y.D.,et al (2015) Ferroelectrics, 475 (1), pp. 135-147 and Hübner, W., Blume, A., et al (2005) International Journal of Artificial Organs, 28 (1), pp. 66-73.
- Please review the reference list again. There is a feeling that the number of old articles does not confirm in any way that the relevance and all new achievements are disclosed properly
Round 2
Reviewer 1 Report
Hydroxyapatite is the main inorganic component of mammalian bones and teeth and is widely used as a therapeutic material for orthopaedic and dental related diseases. This review article reviews recent research work on how to enhance the dispersibility of nano-hydroxyapatite in coated composites to improve the histocompatibility and therapeutic function of the coatings. Overall, it provides a comprehensive overview of the current state of research and also provides useful information for future research work. Therefore, in the current state of the article, I recommend publication.
Author Response
Thank you for reading our manuscript and for giving us a positive evaluation.
Reviewer 2 Report
The revised version did not answer the comments made in the initial review.
Only one section on radiation effect on hydroxyapatite is added.
This is not too relevant to the overall topic.
Author Response
We humbly accept the harsh criticism of our manuscript. In order to broaden the scope of the review, we have added the section “2.1 Chemical bonding of natural substances to the nHA surface to improve the dispersibility” to discuss the chemical bonding of natural organic compounds for HA dispersion. See page 2, 3 and references 21,22,23.
Reviewer 3 Report
Before acceptance, I recommend to replace reference
[18] Bystrov, V.S.; Coutinho, J.; Avakyan, L.A.; Bystrova, A.V.; Paramonova, E.V.; Dekhtyar, Y.D. Computational modeling and studies of hydroxyapatite with defects of the oxygen vacancy type providing its photocatalytic activity. Biosens J. 2018, 7
which is about photocatalytic properties of HAP by another paper of the same authors
Bystrova, A.; Dekhtyar, Y.D.; Popov, A.; Coutinho, J.; Bystrov, V. Modified hydroxyapatite structure and properties: Modeling and synchrotron data analysis of modified hydroxyapatite structure. Ferroelectrics 2015, 475, 135–147.
which is more related to radiation problems !!!
Author Response
Thank you for your kind evaluation. Reference 18 has been replaced with the paper you recommended. See reference 18.